# ATP and nucleic acids competitively modulate LLPS of the SARS-CoV2 nucleocapsid protein

Mei Dang[1], Tongyang Li[1] & Jianxing Song [1✉]

SARS-CoV-2 nucleocapsid (N) protein with very low mutation rates is the only structural protein which not only functions to package viral genomic RNA, but also manipulates host-cell machineries, thus representing a key target for drug development. Recent discovery of its liquid-liquid phase separation (LLPS) opens up a new direction for developing anti-SARS-CoV-2 strategies/drugs. However, so far the high-resolution mechanism of its LLPS still remains unknown. Here by DIC and NMR characterization, we have demonstrated: 1) nucleic acids modulate LLPS by dynamic and multivalent interactions over both folded NTD/CTD and Arg/Lys residues within IDRs; 2) ATP with concentrations > mM in all living cells but absent in viruses not only binds NTD/CTD, but also Arg residues within IDRs with a Kd of 2.8 mM; and 3) ATP dissolves nucleic-acid-induced LLPS by competitively displacing nucleic acid from binding the protein. Our study deciphers that the essential binding of N protein with nucleic acid and its LLPS are targetable by small molecules including ATP, which is emerging as a cellular factor controlling the host-SARS-CoV-2 interaction. Fundamentally, our results imply that the mechanisms of LLPS of IDR-containing proteins mediated by ATP and nucleic acids appear to be highly conserved from human to virus.

[1] Department of Biological Sciences, Faculty of Science, National University of Singapore, 10 Kent Ridge Crescent, 119260 Singapore, Singapore.
✉email: dbssjx@nus.edu.sg

Severe Acute Respiratory Syndrome Coronavirus 2 (SARS-CoV-2) is a member of coronaviruses with ~30 kb genomic RNA (gRNA) packaged by nucleocapsid (N) protein in a membrane-enveloped virion, which caused the ongoing pandemic with >665 millions of infections and >6.69 millions of deaths[1]. SARS-CoV-2 has four structural proteins: namely the spike (S) protein that recognizes the host-cell receptors angiotensin converting enzyme-2 (ACE2), membrane-associated envelope (E), membrane (M) proteins and nucleocapsid N protein[2]. In order to determine their potential for development of vaccine and drug, there is an urgent need to understand structures and functions as well as roles in the viral life cycle of SARS-CoV-2 proteins beyond the spike protein which is currently used for vaccine but undergoes a rapid mutation.

N protein is the only structural protein which not only functions to package gRNA, but is also responsible for suppressing the immune system and manipulating the cellular machineries of the host cell to enhance the viral infection and replication[3–8]. For example, N protein has been currently identified to play critical roles in hijacking host cell machineries for RNA replication and transcription[9,10], nucleocapsid assembly, virion assembly and virion package[11]. Furthermore, N protein provides the connection between viral E, M proteins and gRNA within virion[12]. In particular, it has high immunogenicity[7] and a low rate of mutation, with 91% identity to that of SARS-CoV-1, which is much more conserved than the S protein. Consequently, N protein represents a key candidate for future drug and vaccine development[3–16].

Most strikingly, liquid-liquid phase separation (LLPS), the emerging principle for commonly organizing the membrane-less organelles (MLOs) critical for cellular physiology and pathology[17–19], has been very recently identified as a mechanism underlying the diverse functions of SARS-CoV-2 N protein[20–26]. Noticeably, most functions of N proteins including LLPS are dependent on its binding to various viral/host-cell nucleic acids including single- and double-stranded RNA/DNA of diverse sequences. For example, SARS-CoV-2 N protein has been previously shown to phase separate upon introducing various nucleic acids of specific and non-specific sequences. Moreover, although the detailed mechanism still remains poorly understood, the final package of the RNA genome requires the complex but precise interaction between N protein and gRNA, which should be extremely challenging for SARS-CoV-2 with such a large RNA genome (~30 kb). In this context, any small molecules capable of modulating the interaction of N protein with nucleic acids is anticipated to considerably intervene in key steps of the viral life cycle, some of which might be further developed into anti-SARS-CoV-2 drugs.

ATP, the universal energy currency for all living cells, has cellular concentrations from 2 to 16 mM depending on the types of cells, which are much higher than required for its classic functions[19,27–30]. By contrast, viruses lack the ability to generate ATP[27]. Emerging results indicate that ATP appears to have a novel category of energy-independent functions at mM, which include the modulation of LLPS and specific binding to a list of the nucleic-acid-binding domains of diverse folds at mM[19,28–30]. Very recently we found that ATP is in fact capable of specific binding to the pockets on NTD[25] and CTD[31] with the dissociation constant (Kd) of $3.3 \pm 0.4$ and $1.49 \pm 0.28$ mM as well as dissolving LLPS of SARS-CoV-2 N protein but with the mechanism unknown. These results imply that at mM ATP might not only have functions to control protein hemostasis in living cells, but also act as a key host-cell factor controlling the interaction of the host cell with viruses such as SARS-CoV-2 which are unable to generate ATP.

SARS-CoV-2 N protein is a 419-residue multidomain protein composed of the folded N-terminal domain (NTD) and C-terminal domain (CTD), as well as three long intrinsically-disordered regions (IDRs) containing a number of Arg/Lys residues (Fig. 1a, b and Supplementary Fig. 1a). Previous studies have established that its NTD is a nucleic-acid-binding domain (RBD) functioning to interact with various RNA and DNA of specific and non-specific sequences[13–15], while CTD was previously established to dimerize/oligomerize to form high-order structures[15,16]. Nevertheless, very recently, we found that its CTD is also a cryptic domain for binding ATP and nucleic acids[31]. Very importantly, the sequences of N protein are highly conserved in different variants including Delta and Omicron not only over the folded NTD and CTD, but also over three IDRs (Fig. 1a). This observation implies that IDRs might also have critical functional roles which remain to be discovered.

Despite its criticality for developing therapeutic strategies/molecules, the high-resolution mechanisms for LLPS of N protein still remain unknown. This is most likely due to the challenge in characterizing the aggregation-prone N protein by the high-resolution biophysical methods particularly by NMR spectroscopy, the only available method to experimentally obtain the residue-specific knowledge of LLPS[32–36]. Here, to overcome the challenge, we dissected N protein into differential combinations of domains followed by NMR and DIC characterization on interactions with S2m, a 32-mer stem-loop II nucleic acid motif (Supplementary Fig. 1b) derived from SARS-CoV-2 gRNA as well as its modulation by ATP. Previously RNA, ssDNA and dsDNA forms of SARS-CoV-1 S2m have been shown to bind N protein in the same mode[37]. Furthermore, for SARS-CoV-2, it has been very recently shown that its genome RNA not only could be reverse-transcribed into ssDNA, but also integrated into human genome as double-stranded DNA (dsDNA)[38]. This finding suggests that the different firms of S2m may be the biological ligands of the N protein, which is in general consistent with the notion that many nucleic-acid-binding domains have a very broad spectrum of specificity[28–31,39–42]. For example, human TDP-43 protein containing RNA-binding motif (RRM) domains has been uncovered to functions by binding a large array of RNA and DNA including more than 6000 RNA species of diverse sequences and structures[40–42]. As such, here we used the ssDNA form of S2m because unlike RNA which is prone to degradation by RNAse extensively existing in environments, ssDNA has a very high chemical stability, thus allowing to acquire time-consuming NMR spectra[41,42]. The obtained results decode: 1) N (1-249) is not only amenable for high-resolution NMR studies, but also phase separate upon induction by S2m and modulation by ATP in the same manner as the full-length N protein. 2) ATP binds the nucleic-acid-binding pocket of the folded NTD as well as Arg residues within IDRs in the context of N (1-249) with the average Kd values of $2.0 \pm 0.2$ and $2.8 \pm 0.2$ mM respectively. 3) Residues-specific NMR results reveals that S2m induces and subsequently dissolves LLPS by multivalent binding to both folded NTD and Arg/Lys residues within IDRs. ATP dissolves S2m-induced LLPS by competitively displacing S2m from being bound with the protein. In light of the mechanism of LLPS of human FUS protein we previously revealed[28–30], our present study implies that the mechanisms of LLPS induced by nucleic acids for the proteins composed of both folded nucleic-acid-binding domains and Arg/Lys residues within IDRs are highly conserved from human to virus. ATP is not only emerging as a key cellular factor controlling the host-SARS-CoV-2 interaction, but also provides the rationale/lead for further development of anti-SARS-CoV-2 molecules which can tightly occupy the ATP-binding pockets of N protein to inhibit its interaction with nucleic acids and to disrupt LLPS.

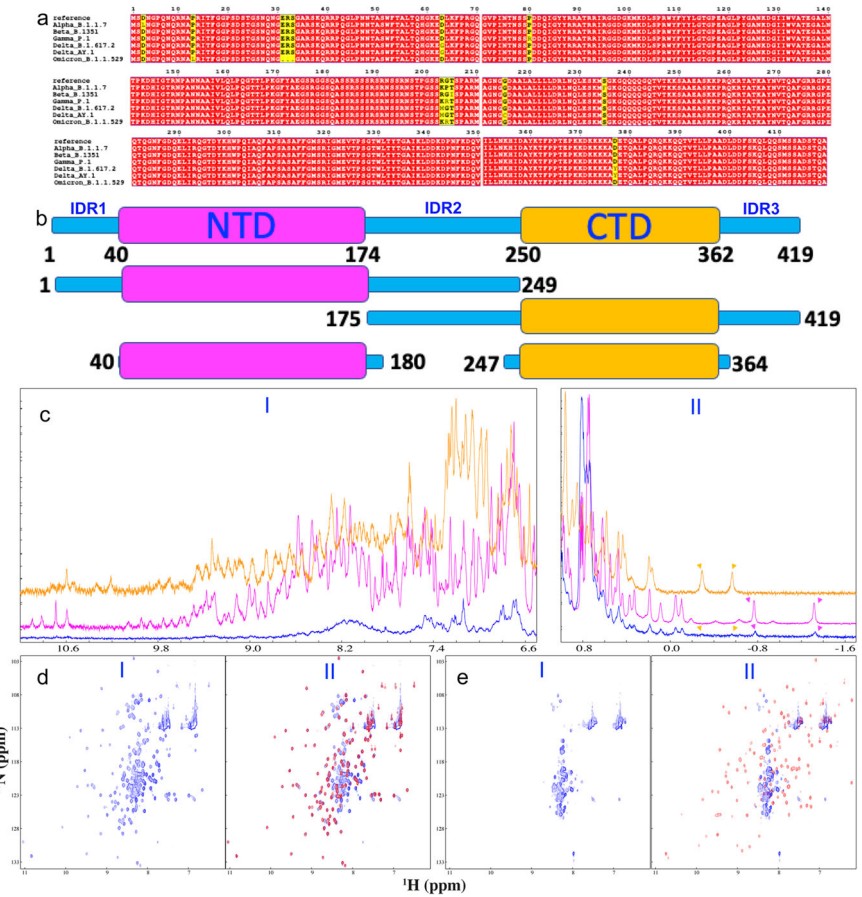

**Fig. 1 NMR characterization of differentially-dissected domains of SARS-CoV-2 N protein. a** Sequence alignment of N protein of Variants of Concern (VOCs) of SARS-CoV-2 according to WHO: https://www.who.int/en/activities/tracking-SARS-CoV-2-variants. **b** Differentially-dissected domains of SARS-CoV-2 N protein used in this study. **c** One dimensional NMR proton spectra of the full-length N protein (blue), NTD (purple) and CTD (light brown) for the amide proton (I) and side-chain (II) regions. Arrows are used to indicate very up-field signature NMR signals of methyl groups of NTD (purple) and CTD (light brown) in 1D NMR spectra. **d** HSQC spectrum of N (1-249) (I) and superimposition of HSQC spectra of N(1-249) (blue) and NTD (40-180) (red) (II). **e** HSQC spectrum of N (175-419) (I) and superimposition of HSQC spectra of N (175-419) (blue) and CTD (247-364) (red) (II).

## Results

**Dissection of N protein and their NMR characterization.**
Initially we attempted to study the full-length N protein. Unfortunately, even after an extensive screening of protein concentrations and buffer conditions, the NMR resonance signals of the full-length N protein were very broad as exemplified by its one-dimensional NMR proton spectra (Fig. 1c), and consequently the signals of its amide protons were too broad to be detected in HSQC spectrum even without phase separation, consistent with a just published NMR study[14].

To identify the domains suitable for high-resolution NMR studies, we first dissected the full-length N protein into two large fragments consisting of differential domains, namely N (1-249) with IDR1, NTD and IDR2 as well as N (175-419) with IDR2, CTD and IDR3, together with the isolated NTD and CTD (Fig. 1b). The isolated NTD and CTD both have well-dispersed HSQC spectra even with peaks highly superimposable to those of the isolated NTD and CTD previously collected in slightly different buffers[13,38,39]. In particular, in their one-dimensional proton NMR spectra (Fig. 1c), NTD has two very up-field signature signals respectively at −0.78 and −1.37 ppm while CTD has two at −0.25 and −0.58 ppm, which are all from the methyl groups with the close contact to the aromatic rings only observed in the well-folded proteins. Strikingly N (1-249) has a well-dispersed HSQC spectrum (I of Fig. 1d) and in particular, the HSQC peaks of its NTD residues are highly superimposable to

those of the isolated NTD residues (II of Fig. 1d), indicating that the structures of NTD are highly similar in both isolated NTD and N (1-249).

By contrast, N (175-419) has a narrowly-dispersed HSQC spectrum (I of Fig. 1e) in which the well-dispersed peaks of the isolated CTD were completely undetected. Nevertheless, a close examination showed that the signature peaks of CTD could be still observed in the 1D spectra of the full-length (II of Fig. 1c) as well as (175-419) (Supplementary Fig. 2), implying that CTD is also similarly folded in the isolated domain, N (175-414) and the full-length N proteins. To confirm this, we further dissected N (175-419) into N (175-364) and N (247-419). As shown by their 1D spectra (Supplementary Fig. 3a), N (175-464) and N (247-419) also have the signature peaks respectively at −0.25 and −0.58 ppm characteristic of the folded CTD which are less broad than those of N (175-419). In particular, well-dispersed HSQC peaks of CTD became detectable for N (175-464) and N (247-419), which are largely superimposable to those of the isolated CTD (Supplementary Fig. 3b, c). As such, most likely due to the self-association/oligomerization or/and μs-ms conformational dynamics provoked by the presence of IDRs and CTD, their NMR signals became broadened to different degrees for N (175-419) and full-length N proteins, consistent with the recent NMR reports[14,36,43,44].

Intriguingly, even in N (1-249) while most of HSQC peaks of the non-proline NTD and IDR1 residues could be detected,

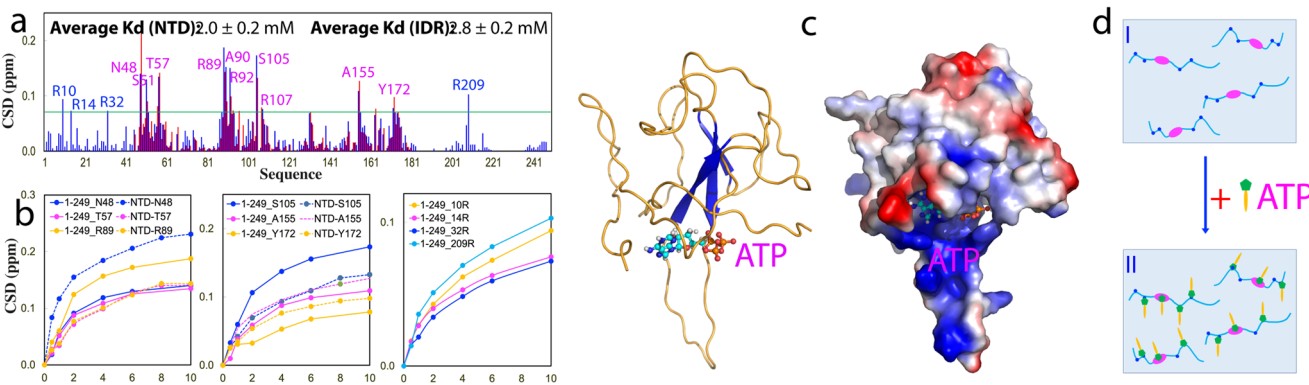

**Fig. 2 Residue-specific view of the ATP binding to NTD and IDR residues of N (1-249). a** Residue-specific chemical shift difference (CSD) of N (1-249) (blue) and isolated NTD (red) between the free state and in the presence of ATP 10 mM. The largely perturbed residues are defined as those with the CSD values at 10 mM > 0.072 (average value + one standard deviation) (cyan line). **b** Shift tracings of HSQC peaks of six representative NTD residue in the context of N (1-249) and isolated NTD, as well as 4 Arg residues within IDRs in the presence of ATP at different concentrations. **c** The docking structure of the ATP-NTD complex we previously constructed for the isolated NTD. **d** Schematic representation of N (1-249) in the free state (I) and in the presence of the exceeding amounts of ATP (II).

HSQC peaks of several segments of IDR2 were too broad to be detected, which include Ser184-Ser197, Leu221-Leu222, and Leu224-G236, thus implying that these segments are involved in µs-ms exchanges/dynamics as recently revealed by NMR backbone dynamics[14,36]. Strikingly, these segments contain several hydrophobic residues Leu and particularly are rich in Ser residues (Fig. 1a) characteristic of the prion-like domains, which, as previously found, are prone to the self-association to form amyloid fibrils[45]. Similarly, a very recent study also showed that CTD with IDRs together led to the disappearance of NMR signals of the full-length N protein although NTD and CTD were shown to have no detectable interaction with each other[14].

**ATP specifically binds residues of both folded NTD and IDRs.** Previously we have found that ATP specifically binds the nucleic-acid-binding pocket of the isolated NTD with HSQC peaks of 11 residues largely perturbed[25]. Here we titrated N (1-249) at the same protein concentration (100 µM) and in the same buffer with ATP at 0.5, 1.0, 2.0, 4.0, 6.0, and 10.0 mM respectively. Upon further increasing ATP concentration, a small set of HSQC peaks underwent large shift as illustrated by Supplementary Fig. 4a. Noticeably, in addition to the NTD residues, several peaks of the residues within IDRs unique for N (1-249) also underwent large shift (Supplementary Fig. 4b, c). Supplementary Fig. 4d presents the expanded shift tracings of two selected residues: Ala90 within NTD and Arg32 from IDR1.

Detailed analysis led to identification of largely-perturbed residue of N (1-249) upon adding ATP. The overall pattern of shifted HSQC peaks of the NTD residues are very similar in both isolated NTD and N (1-249) (Fig. 2a). Precisely, 10 NTD residues have considerable shifts including all those previously identified 11 residues in the isolated NTD except for Leu56, namely Asn48, Ser51, Thr57, Arg89, Ala90, Arg92, Ser105, Arg107, Ala155 and Tyr172 (Fig. 2b). Comparison of the shift tracings of six representative residues showed that in the context of N (1-249) their shift tracings approached being saturated at the slightly lower ATP concentrations than for the isolated NTD (Fig. 2c). Indeed, the fitting of the shift tracings of 10 NTD residues in N (1-249) gave the average Kd of 2.0 ± 0.2 mM (Supplementary Table 1), which is only slightly smaller than that for the isolated NTD (3.3 ± 0.4 mM), which is similar to our previous observations that ATP binds the tandem-linked RRM domains (Fig. 2d) with the slightly higher affinity than the isolated ones of TDP-43

and hnRNPA1[46]. Most strikingly, ATP also induced the large shifts of four Arg residues respectively within IDR1 and IDR2, namely Arg10, Arg14, Arg32, and Arg209 (Fig. 2a). The fitting of their shift tracings (Fig. 2b) gave the average Kd of 2.8 ± 0.2 mM (Supplementary Table 1), which is only slightly larger than that of the NTD residues in the same context of N (1-249).

To exclude the possibility that the observed shifts are due to the presence of $Mg^{2+}$ in complex with ATP, we titrated $MgCl_2$ into N (1-249) and CTD as monitored by HSQC. No large shift was detected for both N (1-249) (Supplementary Fig. 5a) and CTD (Supplementary Fig. 5b) even with $MgCl_2$ concentrations up to 20 mM. The results together reveal that ATP is able to bind the nucleic-acid-binding pocket of the folded NTD in the context of N (1-249) with the complex structure very similar to what we previously constructed for the isolated NTD (Fig. 3c). Furthermore, here we decoded that ATP is also capable of binding Arg residues within IDRs with Kd of 2.8 mM (Fig. 3d), whose affinity is comparable to those for binding a list of the folded nucleic-acid-binding domains recently identified[30,33].

**ATP also binds the folded CTD in N (175-419).** Very recently, we shown that in addition to acting for dimerization/oligomerization, CTD in fact is a cryptic domain for binding ATP and S2m[31]. In particular, CTD binds ATP with Kd of 1.49 ± 0.28 mM[31]. Here we asked the question whether ATP can bind N (175-419). As evidenced by its 1D spectra in the presence of ATP at different concentrations (Supplementary Fig. 6a), ATP did specifically induce the shift of one very up-field NMR peak at -0.25 but not the one at -0.58 ppm, exactly as we previously observed on the isolated CTD[31]. This result clearly indicates that ATP is able to bind the folded CTD in the context of N (175-419). On the other hand, however, as N (175-419) have HSQC spectrum with many peaks undetectable likely due to µs-ms conformational dynamics or/and oligomerization, although ATP also triggered the broadening and disappearance of some HSQC peaks of IDRs (Supplementary Fig. 6b–e), it is impossible to assign these peaks to the corresponding residues.

**S2m modulates LLPS of the full-length N and N (1-249) proteins.** Previously we found that A24, a 24-mer non-specific nucleic acid, was sufficient to achieve the biphasic modulation of LLPS of N protein: induction at low concentrations but dissolution at high concentrations[25]. Here, we selected S2m, a nucleic

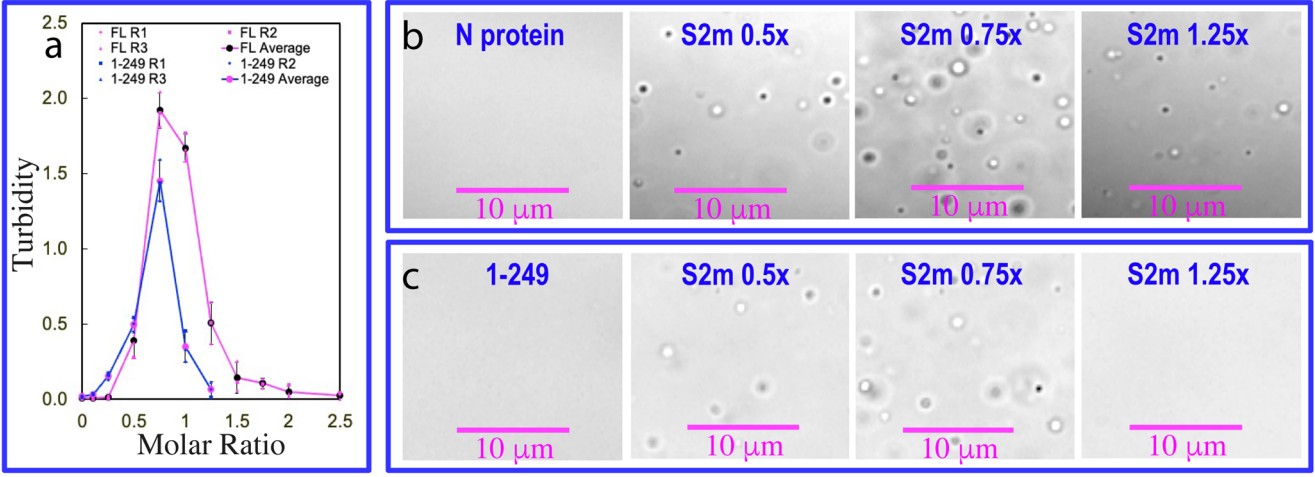

**Fig. 3 S2m biphasically modulates LLPS of the full-length N and N (1-249) proteins. a** Turbidity (absorption at 600 nm) curves of the average values of three repeated measurements ($n = 3$) with STD displayed as error bars of the full-length N and N (1-249) proteins in the presence of S2m at different ratios. **b** DIC images of the full-length N protein in the presence of S2m at different ratios. **c** DIC images of N (1-249) protein in the presence of S2m at different ratios.

acid probe derived from gRNA which was previously utilized to identify nucleic-acid-binding domains of SARS-CoVs[37] to assess its modulation of LLPS in parallel for the isolated NTD and CTD as well as the full-length N and N (1-249) proteins as monitored by measuring the turbidity (absorption at 600 nm) and imaging with DIC microscopy, as we previously conducted on SARS-CoV-2 N protein[25], FUS[28] and TDP-43[29].

We first titrated S2m into the isolated NTD and CTD with protein concentrations reaching up to 200 μM but found no phase separation. By contrast, as shown in Fig. 3, S2m could biphasically modulate LLPS of the full-length N protein: induction at low ratios and dissolution at high ratios. Briefly, the N protein sample showed no LLPS in the free state. However, phase separation was induced upon addition of S2m as evident by the increase of turbidity and DIC imaging. At 1:0.75 (N:S2m), the turbidity reached the highest of 1.92 (Fig. 3a) and many liquid droplets with the diameter of ~1 μm were formed (Fig. 3b). However, further addition of S2m led to the reduction of turbidity and dissolution of the droplets. At 1:1.5 all liquid droplets were completely dissolved.

Subsequently, we titrated S2m into the N (1-249) protein under the same conditions. Again N (1-249) showed no phase separation in the free state. Nevertheless, S2m could also biphasically modulate LLPS of the N (1-249) protein. Briefly, LLPS could be induced upon addition of S2m and at 1:0.75 (1-249:S2m), the turbidity reached the highest of 1.45 (Fig. 3a) and many liquid droplets with the diameter of ~1 μm were also formed (Fig. 3c). Similar to what was observed above on the full-length N protein, further addition of S2m led to the reduction of turbidity and dissolution of the droplets. At 1:1.25 all liquid droplets were dissolved. Compared with the liquid droplets formed by the full-length N protein in the presence of S2m at 1:0.75, the droplets formed by the N (1-249) protein with S2m at 1:0.75 have similar sizes but the number were less, thus resulting in the lower turbidity (Fig. 3a).

**Residue-specific NMR view of LLPS of N (1-249) modulated by S2m.** So far, no high-resolution mechanism has been reported on LLPS of N protein induced by nucleic acid. Here, our successful identification of N (1-249) offered us the opportunity to gain residue-specific view of LLPS by NMR spectroscopy. To achieve this, we monitored the S2m-modulated LLPS by NMR upon stepwise addition of S2m into the N (1-249) sample at ratios

1:0.05, 1:0.1, 1:0.25, 1:0.75, 1:1, and 1:2,5, which range from the induction to complete dissolution of liquid droplets.

As shown in I of Supplementary Fig. 7, even upon addition of S2m at 1:0.05, some HSQC peaks became broadened and consequently their intensity reduced. At 1:0.1 (II of Supplementary Fig. 7) further broadening was observed for many HSQC peaks. Noticeably, at 1:0.25, HSQC peaks of all folded NTD and some IDR residues became disappeared while at 1:0.75, even the intensity of the remaining peaks of IDR residues became largely reduced (III of Supplementary Fig. 7). On the other hand, however, further addition of S2m to 1:1 led to the intensity increase for remaining IDR HSQC peaks, and at 1:2.5, the intensity of those IDR HSQC peaks further increased (V of Supplementary Fig. 7). Nevertheless, even at 1:2.5 where liquid droplets were completely dissolved, the HSQC peaks of NTD and some IDR residues still remained undetectable (VI of Supplementary Fig. 7).

Detailed analysis of the peak intensity of the spectra at different S2m ratios reveals a very interesting picture (Fig. 4): as illustrated by Fig. 4a, even with the addition of S2m at 1:0.05, the intensity of the NTD peaks largely reduced with an average of 0.51 while the intensity of IDR peaks showed the relatively small reduction with an average of 0.79. At 1:0.1, the intensity of the NTD peaks further reduced with an average of 0.31 while the intensity of IDR peaks still has an average of 0.73. Strikingly, at 1:0.25, all NTD peaks became too weak to be detected but IDR peaks still has an average of 0.54. At 1:0.75, the intensity of the remaining IDR peaks became further reduced to 0.34. Intriguingly, however, with further addition of S2m to 1:1, the intensity of IDR residues became increased with the average of 0.48 and further back to 0.76 at 1:2.5.

Furthermore, a close examination indicates that based on the patterns of the intensity changes, the N (1-249) residues can be grouped into three categories (Fig. 4b): the folded NTD residues (group 1) and 7 Arg/Lys residues within IDRs (group 2) with their HSQC peak intensity reduced rapidly upon adding S2m and becoming too weak to be detected above 1:0.25, Interestingly those two categories of residues had their HSQC peaks remaining undetectable even with further addition of S2m up to 1:2.5. By contrast, for other IDR residues (group 3), their intensity reduced with addition of S2m and reached the lowest at 1:0.75. However, with further addition of S2m, their peak intensity increased and at 1:2.5, the average intensity is back to 0.76.

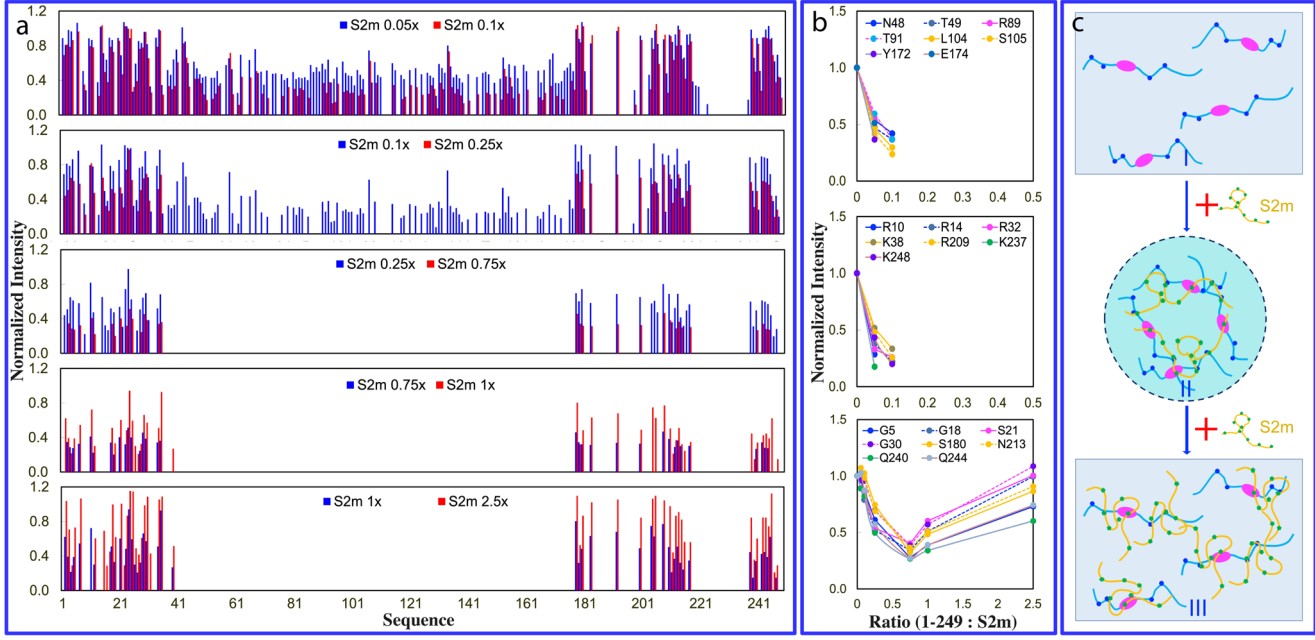

**Fig. 4 Residue-specific view of the biphasic modulation of LLPS of N (1-249) by S2m. a** Normalized intensity of N (1-249) residues in the presence of S2m at different ratios as divided by that of N (1-249) residues without S2m. **b** Normalized intensity of three groups of N (1-249) residues in the presence of S2m at different ratios. **c** Speculative model of homogeneous solution of N (1-249) (I) which is induced to phase separate upon adding S2m at low ratios (II) followed by dissolution into homogeneous solution at high ratios (III).

In titrations with S2m, the intensity of HSQC peaks of N (1-249) appears to be mechanistically modulated by at least two processes: 1) the formation of large and dynamically crosslinked complexes between S2m and N (1-249) molecules, which is expected to slow the rotational motions and consequently lead to the broadening of HSQC peaks; and 2) the binding-induced dynamics on µs-ms time scale that also result in the broadening of HSQC peaks[28,29,33,34,47–49]. As NTD is a folded domain, its residues largely behave as a coupled unit. By contrast, due to the lack of the folded structure, IDR residues behave rather independently. As such, upon binding with S2m, HSQC peaks of all NTD residues become uniformly broadened and disappeared due to the µM Kd or/and provoked µs-ms dynamics. By contrast, IDR residues have rather independent dynamic behaviors and therefore only the peaks of Arg/Lys residues directly bound with S2m become largely broadened and disappeared, while the peaks of other IDR residues only have reduced intensity due to the slowing of the rotational motions upon forming the large dynamically-crosslinked complexes[28,44]. However, upon further addition of S2m, the large complexes are disrupted due to the excessive binding and consequently the intensity of most IDR residues except Arg/Lys become increased. Nevertheless, the peaks of NTD and Arg/Lys peaks remain undetectable even in the exceeding presence of S2m because these residues are still bound with S2m even after the complete dissolution of liquid droplets in the presence of the exceeding amount of S2m.

Previously RNA[50,51] and ssDNA[52] were shown to biphasically modulate LLPS of FUS and TDP-43 respectively but the general mechanisms remain largely elusive. Here the results indicate that S2m appears to achieve both induction and dissolution of LLPS of N (1-249) mainly by specifically binding to the folded NTD and Arg/Lys residues within IDRs. In this context, a speculative model was proposed (Fig. 4c). Briefly, upon adding S2m into the homogenous solution of N (1-249) (I of Fig. 4c) at low ratios, N (1-249) and S2m are capable of dynamically and multivalently interacting with each other over both NTD and IDR Arg/Lys residues to form large and dynamically-crosslinked complexes[28,44,48] which manifest as liquid droplets (II of Fig. 4c).

However, with the exceeding addition of S2m, several S2m molecules become bound with one N (1-249) molecule and consequently the large and dynamically-crosslinked complexes become disrupted, thus manifesting as the dissolution of liquid droplets into homogeneous solution (III of Fig. 4c).

**ATP dissolves LLPS of the full-length N and N (1-249) proteins in the same manner.** The residue-specific NMR results together imply that ATP and S2m might in fact have the highly overlapped binding sites on N (1-249), namely NTD and Arg/Lys of IDRs despite having very different affinities. If this is the case, ATP at the higher concentrations than that of S2m should be able to dissolve the S2m-induced LLPS of N (1-249) by competitively displacing S2m from being bound with the proteins. To verify this mechanism, we prepared the phase separated samples of both full-length N and N (1-249) proteins with the pre-presence of S2m at 1:0.75. Subsequently, ATP was added into the samples in a stepwise manner, as monitored by turbidity (Fig. 5a) and DIC imaging (Fig. 5b and Supplementary Fig. 8).

Indeed, ATP could dissolve LLPS of both full-length N and N (1-249) proteins induced by S2m, although ATP completely dissolved LLPS of N (1-249) at the slightly lower ratio (1:500) than that for dissolving LLPS of the full-length N protein (1:750). This difference is most likely due to the presence of CTD in the full-length N protein which might enhance LLPS by additional involvement in binding S2m or/and dimerization/oligomerization. Briefly, as for N (1-249), at the ratio < 1:25, ATP only has minor effect on LLPS. However, at ratio of 1:250, ATP dissolved liquid droplets as evidenced by the reduction of turbidity and disappearance of many liquid droplets. Strikingly, at 1:500, ATP could completely dissolve liquid droplets of N (1-249) protein induced by S2m. The result clearly suggests that ATP and S2m do interplay in modulating LLPS of the full-length N and N (1-249) proteins in the same manner. To exclude the possibility that the above observation is due to the alteration of the conformation of S2m by ATP-Mg complex, we titrated ATP-Mg complex into a S2m sample as monitored by 1D proton NMR spectroscopy on

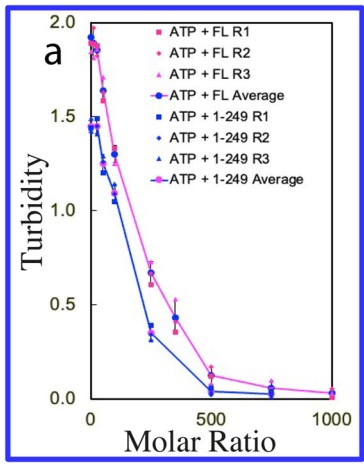
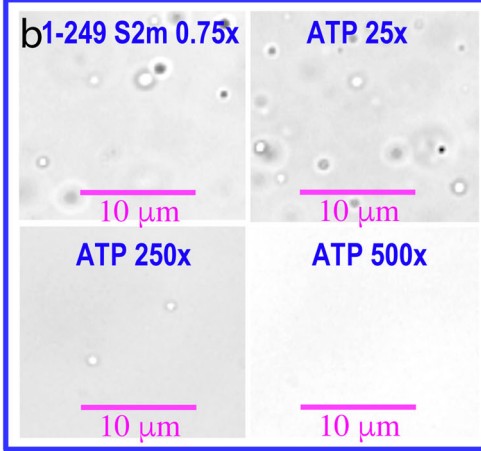

**Fig. 5 ATP dissolves LLPS of N (1-249) induced by S2m. a** Turbidity curves of the average values of three repeated measurements (*n* = 3) with STD displayed as error bars of the full-length N protein and N (1-249) in the presence of S2m at 1:0.75 with additional addition of ATP at different ratios. **b** DIC images of N (1-249) in the presence of S2m at 1:0.75 with additional addition of ATP at different ratios.

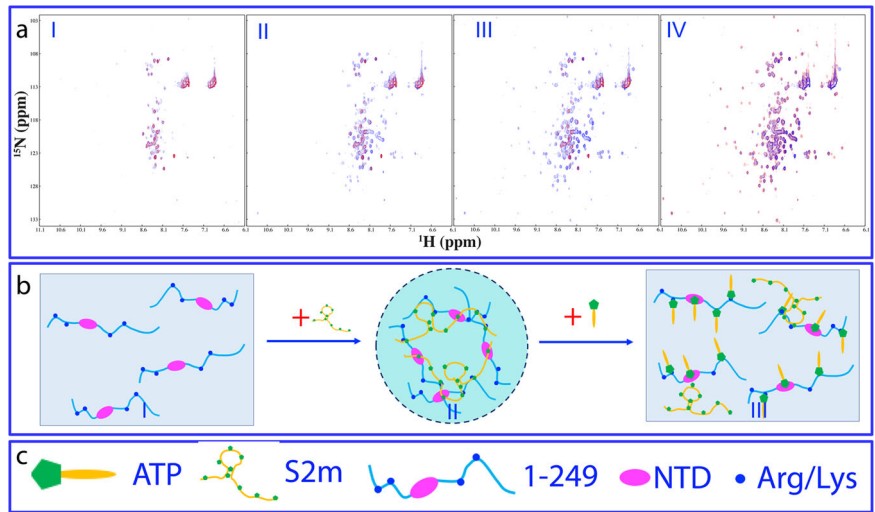

**Fig. 6 NMR view of the interplay of ATP and S2m in modulating LLPS of N (1-249). a** Superimposition of HSQC spectra of N (1-249) in the presence of S2m at 1:0.75 (red) with additional addition of ATP (blue) at 5 mM (I), 10 mM (II) and 20 mM (III). (IV) Superimposition of HSQC spectra of N (1-249) in the presence of ATP at 10 mM only (red) and in the presence of both S2m at 1:0.75 and ATP at 20 mM (blue). **b** Speculative model of N (1-249) in homogeneous solution (I) which undergoes phase separation to form dynamic liquid droplets upon induction by S2m (II), followed by dissolution of droplets into homogeneous solution upon adding the exceeding amount of ATP (III). **c** Illustration icons.

the protons of the base aromatic rings and NH2 which are directly involved in base pairing. The obtained results showed that addition of ATP-Mg complex even up to 10 mM triggered no shift of these NMR signals (Fig. S5C and S5D), thus unambiguously indicating that no conformational alteration occurred for S2m upon addition of ATP-Mg complex.

**ATP dissolves LLPS by competing with S2m for binding the protein.** To understand the high-resolution mechanism, we monitored the dissolution of the S2m-induced LLPS of N (1-249) by ATP with HSQC spectroscopy. As shown in I of Fig. 6a, the addition of ATP at 1 mM into the N (1-249) sample with the pre-existence of S2m at 1:0.75 has very minor effect on its HSQC spectrum. However, addition of ATP at 10 mM led to the restore of some HSQC peaks (II of Fig. 6a) and at 20 mM, the disappeared HSQC peaks including those of the NTD and Arg/Lys residues were re-appeared (III of Fig. 6a). Most strikingly, the HSQC spectrum in the presence of S2m at 1:0.75 with additional addition of ATP at 20 mM is highly similar to that without S2m

but only in the presence of ATP at 10 mM. This observation unambiguously suggests that ATP is able to bind NTD and IDR Arg residues as well as to displace S2m from being bound with the protein. Nevertheless, even at 20 mM, many HSQC peaks, particularly from NTD residues are still weak as compared with those of N (1-249) only in the presence of ATP at 10 mM, implying that ATP even at 20 mM is still unable to completely displace the binding of S2m from N (1-249) because the binding affinity of S2m to N (1-249) is much higher (with Kd of ~μM) than that of ATP (Kd of ~mM).

Here the NMR competition experiments confirm the above speculation that ATP and S2m share the highly-overlapped binding sites over both folded NTD and IDRs of N (1-249). As a consequence, S2m interacts with N (1-249) (I of Fig. 6b) to induce LLPS by forming the large and dynamically crosslinked complexes (II of Fig. 6b), which is characterized by an extensive intensity reduction and disappearance of HSQC peaks. However, the addition of ATP at the high concentrations is able to displace S2m from being bound with N (1-249), thus leading to the

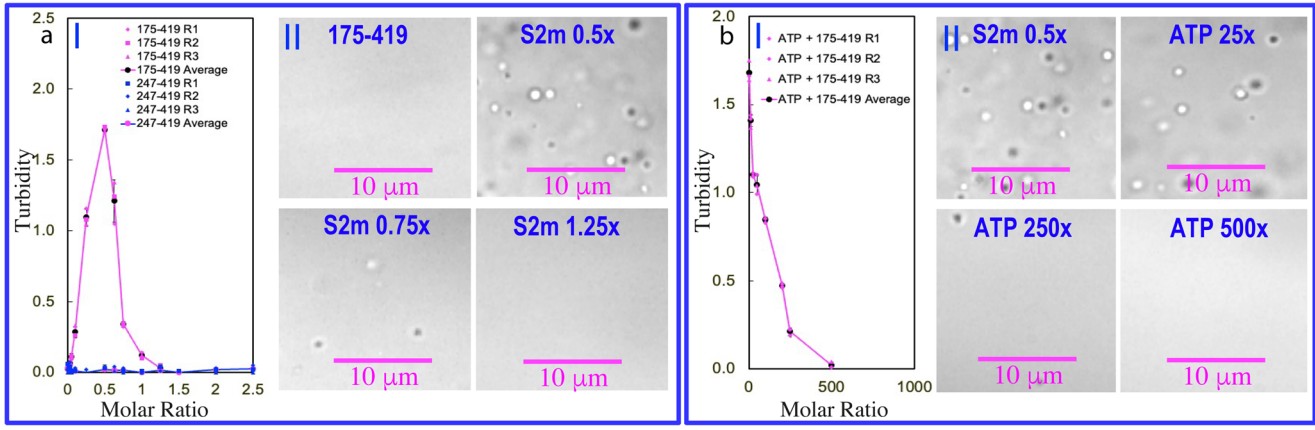

**Fig. 7 S2m and ATP modulate LLPS of N (175-419). a** Turbidity curves of the average values of three repeated measurements ($n = 3$) with STD displayed as error bars of N (175-419) and N (247-419) in the presence of S2m at different ratios (I); and DIC images of N (175-419) in the presence of S2m at different ratios. **b** Turbidity curves of the average values of three repeated measurements ($n = 3$) with STD displayed as error bars of N (175-419) in the presence of S2m at 1:0.5 with additional addition of ATP at different ratios (I); and DIC images of N (175-419) in the presence of S2m at 1:0.5 with additional addition of ATP at different ratios (II).

dissolution of the liquid droplets (III of Fig. 6b), with the reappearance of HSQC peaks including those of the NTD and Arg/Lys residues. Nevertheless, due to the fact that S2m binds N (1-249) with Kd of ~μM, which is much higher that that of ATP (Kd of ~mM), ATP at 20 mM is still unable to completely displace S2m from binding N (1-249), and consequently the intensity of HSQC peaks of N (1-249) in the presence of both S2m and ATP is still weaker than that only in the presence of ATP.

**S2m modulates LLPS of N (175-419) protein.** We also assess whether S2m could induce LLPS for N (175-419), N (175-365) and N (247-419). As shown in Fig. 7a, the N (175-419) sample showed no LLPS in the free state but phase separation was induced upon addition of S2m as evident by the increase of turbidity and DIC imaging. At 1:0.5, the turbidity reached the highest of 1.71 (I of Fig. 7a) and many liquid droplets with the diameter of ~1 μm were formed (II of Fig. 7a). However, as observed on N protein and N (1-249) above, further addition of S2m led to the reduction of turbidity and dissolution of the droplets. At 1:1.25 all liquid droplets were completely dissolved.

On the other hand, for N (175-365), addition of S2m even at 1:0.1 led to visible precipitation and consequently no DIC characterization could be performed. For N (247-419), addition of S2m even up to 1:2.5 induced no LLPS as well as precipitation as reported by turbidity (I of Fig. 7a). The results suggest that CTD needs the additional presence of IDR2 with 8 Arg and 3 Lys residues to achieve multivalent and dynamic binding with S2m to drive LLPS. By contrast, as IDR3 contains only 1 Arg and 9 Lys, the interaction of S2m with Arg/Lys is not sufficiently strong for driving LLPS as the interaction between nucleic acids and Lys is much weaker that that with Arg[23,38].

We also prepared the phase separated samples of N (175-419) with the pre-presence of S2m at 1:0.5. Subsequently, ATP was added into the samples in a stepwise manner, as monitored by turbidity (I of Fig. 7b) and DIC imaging (II of Fig. 7b). Indeed, ATP could dissolve its LLPS induced by S2m and the droplets were completely dissolved at 1:500, which is in general similar to what was observed on N (1-249) and full-length N protein.

**NMR characterization of the interaction of N (247-419) and S2m.** We further set to characterize the interactions of S2m with N (175-419), N (175-365) and N (247-419) by NMR. Unfortunately, as to collect good quality NMR spectra needs a protein

concentration of at least 50 μM, we were unable to collect NMR spectra of (175-419) and N (175-365) as they became precipitated at 50 μM upon adding S2m even at 1:0.1. Nevertheless, we have successfully collected NMR spectra of N (247-419) at 50 μM in the presence of S2m up to 1:2.5 (Supplementary Fig. 9). Intriguingly, although the addition of S2m triggered no LLPS, S2m was shown by NMR (Supplementary Fig. 9) to interact with the folded CTD as evidenced by the broadening of several very up-field NMR signals, exactly as we previously observed on the interaction of S2m with the isolated CTD[31]. On the other hand, as judged by its HSQC spectra in the presence of S2m at different concentrations, most HSQC peaks of the IDR3 residues in N (247-419) retained even up to 1:2.5, implying that the interaction of S2m with IDR3 is indeed relatively weak as compared to that between S2m and Arg-rich IDR1 and IDR2 which resulted in dramatic disappearance of many of their HSQC peaks (Figs. 3, 4).

## Discussion

Recently accumulated results suggest that condensates/compartments of proteins and nucleic acids have widely formed in infected cells by viruses more than SARS-CoV-2, which include HIV, influenza, Measles, Hendra and vesicular stomatitis, Borna disease and rabies viruses. These condensates/compartments are organized by LLPS and also known as viral factories, viroplasm, or viral replication centers, which function in virus-host interaction and viral life cycle including genome replication, gene expression and antiviral immune response[53–61]. Intriguingly, viral proteins are rich in IDRs which have been extensively identified to phase separate upon interacting with the viral or/and host RNA/DNA[61]. Therefore, the delineation of the critical roles of viral LLPS not only sheds light on previously-unknown principles underlying the host-virus interaction and viral life cycle, but most importantly opens up a new direction for development of anti-viral strategies/drugs although this still remains almost completely unexplored so far.

In the present study, we aimed to decipher the high-resolution mechanism for LLPS of SARS-CoV-2 N protein by NMR. The obtained results decode that S2m induces and then dissolves LLPS of N protein by dynamic and multivalent interactions over both folded NTD/CTD and Arg/Lys residues within IDRs. In particular, our studies provide the residue-specific evidence that Arg/Lys residues within IDRs are also capable of directly binding with nucleic acids. As such, in addition to passively providing the

essential dynamics for forming liquid-like droplets, IDRs in fact directly drive LLPS by utilizing their Arg/Lys residues to dynamically interact with nucleic acids. Therefore, LLPS of SARS-CoV-2 N protein is driven by interactions of nucleic acids dynamically and multivalently with its folded NTD/CTD as well as IDRs. Furthermore, our results unveiled that three distinctive scenarios exist for the binding of ATP to the folded domains such as NTD/CTD and IDRs. Briefly, for the folded domains with no capacity in binding nucleic acids, ATP only has very weak and non-specific interactions with them, as we previously observed on human profilin-1[29] and the other group showed on a list of folded proteins[62]. By contrast, for the nucleic-acid-binding domains[30] such as FUS RRM domain[29,33] and NTD/CTD of SARS-CoV-2 N protein, ATP specifically binds to the pockets within their nucleic-acid-binding interface, which is formed only upon the correct folding by diverse residues located over the entire sequence. On the other hand, for IDRs ATP and nucleic acid were both shown to bind Arg/Lys residues in the residue-specific manner by establishing electrostatic interactions between tri-phosphate group pf ATP or phosphate group of nucleic acid and side chain cations of Arg/Lys as well as π-π/π-cation interactions between base aromatic rings and Arg/Lys side chains, as we experimentally found on FUS[28] and TDP-43 PLD[35,41] as well as very recently uncovered by combining quantum mechanical method, mean-field theory and molecular simulations[63].

Previously, RG-/RGG-motifs within IDRs have been identified to bind nucleic acids without a strict requirement of types and sequences[28,64–66]. Here our NMR results showed that none of four IDR Arg residues of N protein which are capable of interacting with nucleic acid are within the RG-/RGG-motifs but instead with very diverse sequences: Gln9-Arg10-Asn11; Pro13-Arg14-Ile15; Glu31-Arg32-Ser33; and Ala208-Arg209-Met210. Therefore, our results extend the previous notion by showing that as long as Arg residue is located within IDRs and accessible, it is capable of interacting nucleic acids without needing to be within RG-/RGG motifs. On the other hand, Lys38, Lys237 and Lys248 were also identified to interact with nucleic acid, in general consistent with the previous report that Lys residue is also critical for interacting with RNA to drive LLPS of the human proteins in P-body[67,68]. However, the binding affinity of nucleic acids to Arg appears to be much higher than that to Lys because the aromatic base rings of nucleic acids can only establish π-cation interaction with the side chain of Lys residue, but π-π and π-cation interactions with the side chain of Arg residue[28,48]. As such, N (247-419) with 6 Lys but only 1 Arg in IDR3 has no capacity in phase separation although S2m is still able to bind the folded CTD.

Most unexpectedly, ATP with concentrations > mM in all living cells but totally absent in viruses, has been previously decoded to bind both the folded NTD and CTD with the Kd of 3.3 ± 0.4 and 1.49 ± 0.28 mM respectively. Here we revealed that Arg residues within IDRs of a viral protein have the comparable affinity with Kd of 2.8 ± 0.2 mM. Most critically, ATP acts to dissolve the S2m-induced liquid droplets by displacing S2m from being bound with the protein to disrupt the large multivalently and dynamically-crosslinked S2m-protein complex, thus clearly indicating that ATP and S2m share the highly overlapped sites over both folded NTD and IDRs of the viral N protein, in generally similar to what we previously observed on human FUS protein[28,29]. Nevertheless, the much higher concentration of ATP is needed to dissolve the S2m-induced LLPS of N protein, because unlike ATP with only one aromatic base ring and one tripho-sphate group, the nucleic acid fragment contains multiple flexibly-linked aromatic base rings and phosphate groups. Consequently, a nucleic acid fragment is sufficient to establish the multivalent binding with the folded NTD or/and IDRs of N protein, thus leading to the much higher affinity[63]. Nevertheless,

the capacity of ATP in dissolving the S2m-induced LLPS of N protein bears the immediate biological relevance because in the host cell ATP always has > mM concentrations but at the initial stage of infection, the number of viral gRNA and N protein should be very low based on the estimation that in one SARS-CoV-2 virion, ~730–2200 copies of N protein form the complex with one 30-kb gRNA[69]. As such, upon immediate entry into the host cell of the virus, ATP is expected to facilitate the uncoating of the SARS-CoV-2 gRNA-N-protein condensates which is essential for initiating the viral life cycle in the host cell. In this sense, ATP appears to be evolutionarily hijacked by SARS-CoV-2 to promote its own life cycle in the host cell and thus emerges a key cellular small molecule which critically controls the host-SARS-CoV-2 interaction.

Pandemic-relevantly, the sequences of SARS-CoV-2 N protein of different variants are highly conserved not only over the folded NTD and CTD, but also over Arg/Lys residues within IDRs. This implies that the mechanisms underlying LLPS of N protein as induced by nucleic acids and dissolved by ATP might be highly conserved in different variants including Delta and Omicron. Therefore, ATP could serve as a key lead for further design of anti-SARS-CoV-2 drugs efficient for different variants. Briefly, small molecules which can occupy the ATP-binding pockets of N protein but with much higher affinity are expected to have anti-SARS-CoV-2 activity because they can block the essential binding of N protein with nucleic acids, as well as disrupt its LLPS. Interestingly, Imatinib, one of three drug candidates WHO was set for clinical trial to treat SARS-CoV-2, is an ATP mimetic originally used to treat cancer by inhibiting tyrosine kinases[70]. However, its mechanism to inhibit SARS-CoV-2 remains completely unknown and therefore it would be of critical interest to investigate whether it can also modulate LLPS of SARS-CoV-2 N protein. As the properties of SARS-CoV-2 N protein required for interacting with nucleic acids to drive LLPS can be extensively found in the IDR-rich proteins of other viruses, it is thus tempting to speculate the mechanisms of LLPS deciphered here as specifically induced by nucleic acids and dissolved by ATP might be also conserved to a certain degree in other viruses. If this is confirmed, it is possible that ATP and ATP-like small molecules might have the common ability to modulate the viral LLPS essential for their life cycle and most important could serve as leads for design of antiviral drugs for other viruses.

## Methods

**Preparation of recombinant SARS-CoV-2 N protein as well as its dissected domains**. The gene encoding 419-residue SARS-CoV-2 N protein was purchased from a local company (Bio Basic Asia Pacific Pte Ltd), which was cloned into an expression vector pET-28a with a TEV protease cleavage site between N protein and N-terminal 6xHis-SUMO tag used to enhance the solubility[25]. The DNA fragments encoding its N (1-249), N (175-419), N (175-364) and N (247-419) as well as NTD (40-180) and CTD (247-364) were subsequently generated by PCR rection and subcloned into the same vector.

The recombinant full-length N protein and its dissected domains were expressed in *E. coli* cells BL21 with IPTG induction at 18 °C, which were found to be soluble in the supernatant. For NMR studies, the bacteria were grown in M9 medium with addition of ($^{15}$NH$_4$)$_2$SO$_4$ for $^{15}$N-labeling. The recombinant proteins were first purified by Ni$^{2+}$-affinity column (Novagen) under native conditions and subsequently in-gel cleavage by TEV protease was conducted. The eluted fractions containing the recombinant proteins were further purified by FPLC chromatography system with a Superdex-200 column for the full-length, N (1-249), N (175-419), N (175-364) and N (247-419) as well as a Superdex-75 column for NTD and CTD[10]. The purity of the recombinant proteins was checked by SDS-PAGE gels and NMR assignments. ATP was purchased from Merck. Protein concentration was determined by spectroscopic method in the presence of 8 M urea[71].

**NMR characterizations of differentially-dissected domains**. Due to the proneness of self-association, after extensive screening of protein concentrations and buffer conditions, the samples of the full-length N and N (175-419) proteins were prepared in 25 mM HEPES buffer (pH 6.5) with 70 mM KCl, while the samples of the other constructs were in 10 mM sodium phosphate buffer (pH 6.5) in the presence of

150 mM NaCl, which mimics the cellular environments. NMR experiments were conducted at 25 °C on an 800 MHz Bruker Avance spectrometer equipped with pulse field gradient units and a shielded cryoprobe as described previously. The proton chemical shift was calibrated with reference to DSS and water[28–31].

**NMR characterizations of the binding of ATP and S2m to N (1-249) and NTD**. NMR samples of N (1-249), N (175-419), and NTD were prepared at 100 μM in 10 mM sodium phosphate buffer (pH 6.5) in the presence of 150 mM NaCl. ATP was dissolved in the same buffer[10,18,20,21,27]. The final solution pH was adjusted to 6.5 by use of very diluted HCl or NaOH.

For NMR titrations to determine residue-specific Kd of N (1-249) residues for binding ATP, two dimensional $^1$H-$^{15}$N NMR HSQC spectra were collected on the $^{15}$N-labelled sample at 100 μM in the presence of ATP at 0, 0.5, 1, 2, 4, 6, 8, 10 mM. For NMR characterization of the binding of N (1-249) with S2m, HSQC spectra were collected on the $^{15}$N-labelled sample at 100 μM with addition of S2m at 0, 1:0.05, 1:0.1, 1:0.25, 1:0.75, 1:1, 1:2.5 (1-249:S2m). For NMR characterization of the binding of N (175-419), N (175-364) and N (247-419) with S2m, HSQC spectra were collected on the $^{15}$N-labelled sample at 50 μM. For NMR investigation on the interplay of ATP and S2m in modulating LLPS of N (1-249), HSQC spectra were collected on the phase separated sample of the $^{15}$N-labelled N (1-249) at 100 μM in the pre-existence of S2m at 1:0.75, into which ATP was subsequently added at different combinations. NMR data were processed with NMRPipe[72] and analyzed with NMRView[73].

**Calculation of CSD and data fitting**. Sequential assignments were achieved based on the deposited NMR chemical shifts for NTD and N (1-249)[38,39]. To calculate chemical shift difference (CSD), HSQC spectra collected without and with ATP or S2m at different concentrations were superimposed. Subsequently, the shifted HSQC peaks were identified and further assigned to the corresponding residues. The chemical shift difference (CSD) was calculated by an integrated index with the following formula:[47]

$$CSD = ((\Delta^1 H)^2 + (\Delta^{15}N)^2/4)^{1/2} \qquad (1)$$

In order to obtain residue-specific dissociation constant (Kd), we fitted the shift tracings with significant shifts (CSD > average + STD) by using the following formula:[10,21,34]

$$CSD_{obs} = CSD_{max}\{([P] + [L] + Kd) - [([P] + [L] + Kd)^2 - 4[P][L]]^{1/2}\}/2[P] \qquad (2)$$

Here, [P] and [L] are molar concentrations of CTD and ligands (ATP) respectively.

**LLPS imaged by differential interference contrast (DIC) microscopy**. The formation of liquid droplets was imaged on 50 μl of the full-length N protein or N (1-249) samples by DIC microscopy (OLYMPUS IX73 Inverted Microscope System with OLYMPUS DP74 Color Camera) as previously described[25]. The protein samples were prepared at 20 μM in 25 mM HEPES buffer (pH 6.5) with 70 mM KCl. The turbidity (absorption at 600 nm) were measured for all DIC samples with three repeats.

**Statistics and reproducibility**. For NMR and DIC experiments, the exploratory experiments of the SARS-CoV-2 N protein and its dissected fragments titrated with ATP and S2m ssDNAs at different concentrations were first conducted to identify the optimized concentration ranges. Subsequently, the final DIC and HSQC titrations were performed once with the optimized points of ATP and S2m ssDNAs concentrations.

**Reporting Summary**. Further information on research design is available in the Nature Research Reporting Summary linked to this article.

## Data availability

The data supporting the findings of this study are available within the paper and Supplementary Data 1. All other data are available from the corresponding author upon reasonable request.

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

## Acknowledgements

This study is supported by Ministry of Education of Singapore (MOE) Tier 1 Grants A-8000711-00-00, R-154-000-B45-114 and R-154-000-B92-114 to Jianxing Song.

## Author contributions

Conceived the research: J.S. Performed research and analyzed data: M.D, T.L. and J.S; Acquired funding: J.S; Wrote manuscript: J.S and M.D.

## Competing interests

The authors declare no competing interests.
