## [Peer Review File · Communications Biology]

Reviewers' comments:

Reviewer #1 (Remarks to the Author):

The report by Dang et al. on the role of ATP and nucleic acid in LLPS of SARS-CoV-2 N protein is potentially of interest. They showed the residue specific interaction of nucleic acid with N protein which is replaced by the same sort of interaction by ATP is certainly of interest but lacks justification in many aspects. Their data of binding are robust, but nothing strongly conclude anything in the aspect of LLPS and there are several major issues which needs to be addressed.

In general, the paper is not well presented, is not easy to follow and presents several experimental limitations as well as the controls. The authors show many NMR experiments that are poorly explained, and figures are not at all presented well. It is very hard from a reader's point of view to understand the chemical shift changes as most of the data are overlapping. It will be always helpful to mark the changes specifically with the arrows along with contrasting colors for overlapping data.

I hope that the suggestions below will help the authors to improve their manuscript.

Especially, figures do not have any specific font size, in some places it's a mixture of smaller and larger font size. Hence, it looks very cluttered. Authors need to create the balance of font size throughout all the figures in order to make it easy for the readers.

Some specific points to improve the manuscripts are-

1) Abstract: "ATP is not only emerging as a cellular factor controlling the host-SARS-CoV-2 interaction, but also provides a lead for developing anti-SARS-CoV-2 drugs efficient for different variants of SARS-CoV-2."

This comment is way too forward compared to the extent of the work. This work is mainly based on NMR and DIC. Whether NMR can efficiently study all the interactions and alterations, DIC is not proven the most efficient method to characterize a LLPS. Authors' approach to quantitatively detect the residue-specific interaction through NMR is exciting, but it cannot be established as lead for drug at this time. For this, it was needed some amount of cellular work to prove the effect of ATP in condensate formation N protein with at least an over-expression construct. Also, there should have more microscopic and other experiments in order to prove LLPS formation and nature of liquid droplets in presence and absence of nucleic acid and ATP.

2) Introduction: Line 102, "energy-independently" should be written as "energy-independent".

3) Line 105, "specifically" should be written as "specific".

4) Line 114, Here it is represented as "SFig 1A" where in line 132, it is as "Fig S1B". Please make the pattern same throughout the whole manuscript.

5) Line 122-127, "Despite the criticality of understanding LLPS of N protein for developing therapeutic strategies/molecules to fight the pandemic, the high-resolution mechanisms for LLPS of N protein still remain unknown most likely due to the challenge in characterizing the association-prone N protein by the high-resolution biophysical methods particularly by NMR spectroscopy, the only one available so far to experimentally obtain the residue-specific knowledge of LLPS (12,21-28)."

This sentence is way too long. Please break it down and make it easy for the understanding of readers.

6) Results: Line 182, Fig 1C, 1D X and Y- scale font sizes are very different, hard to understand. Superimposable images are hard to understand, please use arrows to indicate specific changes in specific residues. Author can use specific color arrows for specific residue changes.

7) In Figure 1C, The comparison of the 1D spectra of different variants of N protein is missing a few

important points. Specifically,

- a) Chemical shift referencing with standard sample is recommended for accurate measurement of the methyl chemical shifts.
- b) The whole region of the sidechain is missing from the comparison.
- c) 1D spectra of the buffers must be added as a control spectra.

8) Line 184-185, "NTD are highly similar in both isolated NTD and N (1-249)"- Authors should add a comparison of 1H-13C HSQC spectra, importantly around the methyl proton region.

9) Line 190, "CTD is also similarly folded in the isolated domain" There is no significant experimental evidence found for this statement. Both the 1D and 2D spectra are significantly different in N (175-419) and CTD (247-364). It looks like N (175-419) is unstructured compared to only CTD (247-364) construct. Also, as the author shows the 1D spectra of the CTD constructs are similar (Fig S3A), it is recommended to show a 1H-13C HSQC comparison.

10) Line 194, "Well dispersed peaks", Can the authors assign those well-dispersed peaks to know the residues?

11) Line 218, Authors have justified their use of high ATP concentration as it is physiologically relevant. But they have use protein N concentration around 100 μ M or so. Please shed light on the concentration of N protein physiologically and talk a little bit about the stoichiometry of protein and ATP in their experiments.

12) Line 221, "HSQC peaks underwent significant shift as illustrated by I of Fig. 2", Superimposition of HSQC spectra of 6 different concentrations of ATP will be useful to track ATP dependent shift of the amide peaks.

13) In III of Fig 2, The peaks in the IDRs become resolved upon 10 mM ATP binding. This signifies significant structural change in IDRs. The authors should add more insights on this.

14) Line 227-235, Authors state that residues with significant shift have CSD > 0.072. However, in their dissociation constant calculation they select only 10 NTD residues. Surprisingly, lots of residues in NTD with CSD > 0.072 were not included in the calculation. How does they affect the dissociation constant?

15) In the legend of "II of Figure 2", isolated NTD (Green) is not mentioned.

16) Line 232, it will be good to shed the light on the reason of exclusion of Ala90 in this work.

17) There was no clear difference in the 1D spectra of free and ATP-bound states in Fig. S4A.

18) Authors' explanation on the involvement of Arg residues are interesting, but there are many reports on binding to Arginine leads to folding of IDR, which may lead to aggregation/oligomerization. Please shed light on this fact how oligomerization and LLPS are controlled.

19) Line 283, Continuing from the previous point, turbidity is also an indication of aggregation. Can author provide some references on turbidity as a measurement of LLPS?

20) Line 283, DIC imaging is not the best way to confirm LLPS. Microscopy with labeled protein can confirm the phase separation. Some more works are suggested in order to characterize LLPS. Some videos on fusion of the liquid droplets are handy along with FRAP to show the nature of LLPS.

21) A major question is, if using high concentration of nucleic acid (S2m) itself is capable of dissolve LLPS, then what is the need of ATP here to replace the interactions. In this case, adding more nucleic acid to the nucleic binding sites would help.

22) Line 341-343, "provoked μ s-ms dynamics", "IDR residues have rather independent dynamic behaviors"

Line 345-346, "the large dynamically-crosslinked complexes"- Can author provide significant references on these comments?

23) In Figure 6C, it looks like their representation after dissociation of LLPS does not support their data. Their data shows, after dissociation there is only existence of binding to IDRs, which is not specifically shown in the figure.

24) Line 377, Both 1:500 and 1:750 is very high. Please shed light on the desired physiological stoichiometry.

25) Line 403, What does it mean by "significant increase of solution viscosity"? How is it measured? Is there any reference to this point?

26) Line 413, "ATP at 20 mM", The NMR measurements was \sim 2.0 mM. Is it a typo?

27) Line 646, Is it CTD or NTD?

Reviewer #2 (Remarks to the Author):

This manuscript by Dang et al. uses an NMR-centered approach to identify mechanistic steps of LLPS caused by the SARS-CoV-2 N protein with RNA, exemplified by the 3'-UTR S2m element. This essential process is thought to be a key event for multiple downstream viral processes, e.g. genome transcription, replication and packaging, all of which involve N as a/the central protein component. The authors are able to provide atom/residue-resolved information for major parts of N and identify the IDRs 1 and 2 as well as both folded domains to be involved in LLPS, while the four also shown susceptibility to ATP, which competes for binding sites in N, in particular the NTD and positively charged residues within IDRs. All this is corroborated by DIC experiments, allowing to quantify LLPS with a second method. In the context of ATP-availability the study suggests that SARS-CoV-2 hijacks the host's ATP metabolism to steer RNA-protein interactions, or that ATP acts as a potential antiviral lead. While major parts of the herein shown is based on a previous publication by the authors (showing ATP-binding by NTD and CTD), the authors here specifically address the role of IDRs, including all parts of N as well as the competition with RNA-binding.

The presented results are a valuable route to follow, both from a pharmaceutical as much as mechanistic point of view, seeing that in fact there are nucleoside/nucleotide analogues being applied, but targeting the RdRP. With this, the study per se represents a strong contribution to the general understanding of host-virus interactions. The data are trustworthy and NMR-DIC experiments do back-up each other and I do find the mechanistic findings convincing.

However, I have a number of major concerns that will need to be addressed before this manuscript is suitable for publication. Not only some findings will need additional, confirmative and control experiments, but also formal points will need to be looked at properly.

In detail, I ask the authors to address the following regarding experiments/discussion/interpretation:

1) Assuming ATP to have a strong physiological role in SARS-CoV-2 LLPS, what about its cofactor Mg²⁺? Could authors provide evidence for effects of magnesium concentration taking into account typical cellular concentrations?

Along with that, authors will need to look at the effects of ATP (and Mg) on the secondary structure of S2m as it might be an indirect effect (see later comment on RNAs) for the LLPS phenotype.

2) Seeing the high concentrations necessary to titrate N protein, I wonder about the stability of pH in NMR samples (knowing about N's susceptibility to changes in pH). Has pH been monitored and can potential effects on CSDs and/or LLPS be separated from ATP?

3) Have the authors considered to include a construct of NTD-IDR2-CTD? According to very recent work (Bessa et al., *Sci Adv.*, 2021) this should be NMR-suitable and would combine the features of specific binding by the NTD, dimerization by the CTD and RNA-binding through IDR2 together with both folded domains binding ATP? Please comment at least!

4) In order to more precisely discuss the role of host ATP for the viral life cycle: I assume there meanwhile is sufficient literature available on proteome changes after SARS-CoV-2 infection, which might give hints on differential levels of ATP synthetases/ATPases? Could the authors comment on that?

5) What is the rationale of using S2m here, it is not the bona fide target of N in SARS-CoV-2 (or is there a reference missing? the reference to the own previous paper does not really help)!? Would one not expect to see even more drastic effects with larger RNAs? Please compare findings (in DIC) using a short and long (ideally more/less structured) RNA. Would one need a much higher excess of ATP to dissolve LLPS with large RNAs?

I do not demand a full experimental setup considering all parameters like N oligomeric states and the

role of RNA-structural content. But it appears too selective to only look at one RNA only. I also did not understand, if CTD and NTD on their own bind to S2m? What are the affinities then?

6) Where was S2m taken from or how was it produced? This is not given in the Methods section.

7) I am puzzled by the usage of the term specificity in the manuscript. It might not be correct or actually needed in all places where used. E.g. what is the "specific" binding of ATP to NTD and IDRs? Isn't it almost all Args in the end? This term "specific" needs to be discussed. (see also page 24, line 564) In particular for the NTD, how can it be specific if only the hinge region of the hairpin is affected and not the hairpin residues themselves (which have been shown to interact with RNA "specifically")? Is this meant with specific? And if ATP is not bound by all positive residues, how would that be explained within IDRs? Please discuss!

Along the same route: page 16, line 406: why does S2m "specifically" interact with 1-249? Compared to other RNAs or proteins? What is meant here, since N will bind lots of other RNAs from SARS-CoV-2 as well!

8) On affinities: How is the higher affinity for 1-249 for ATP explained when looking at the NTD peaks? Should those not be in competition with binding sites in IDRs? (page 9, bottom statement). This does not completely convince me in its current description. Have different oligomer states between constructs (also important for interpreting LLPS) been taken into consideration? In fact, I do not agree that NTD affinity for ATP is higher in N1-249, looking at the provided spectra. Maybe the authors could corroborate their calculations and statements with corresponding raw data (e.g., seeing the two peaks between 9.35 and 9.5 / 121 and 123).

9) Regarding NMR titrations: why would some peaks not reappear even at high excess of RNA? (page 13, top) Please discuss in a bit more detail (exchange?)! I also suggest to include an intensity plot of free vs. recovered spectra after LLPS instead of just mentioning it along with full spectral views? It should be shown as those data are available.

10) On Figures: Figures should be designed/labeled (inset text) to express what is claimed. For some of them, I suggest to provide additional zoom-ins rather than sole full-spectral views. This includes the Kd determination: Exemplary shifting residues should be shown indicating the data quality (seeing the minor CSPs present at maximal ATP concentration). In its present form the large number of HSQCs hardly makes a link to the derived data.

11) I strongly suggest shortening of the manuscript. It appears very redundant in information throughout the intro, results and discussion and readability will improve by removing e.g. too much interpretation in the intro. Also the language style seems to shift in between (which is ok for me), and there are many incomplete sentences (see minor issues for examples).

12) There is insufficient handling of available literature and referencing per se. E.g., the second paragraph of the introduction cites 15 references in a block, while e.g. line 80 misses a good point to more specifically put a reference. Please be more precise here in what references refer to what finding! The same for page 3, line 88: I do not agree that those citations claim LLPS to be a key mechanism for the diverse viral functions, although it might be a strong indication. Also: page 4, line 92: please add a reference!

Minor points are:

- I find the title misleading. Unified to or with what?
- inconsistency in labeling in Fig. 1: NTD 40-180 vs. legend NTD 44-180
- page 3, line 67: numbers of infections and deaths based from SARS-CoV-2 should be referenced with a particular day looked at
- page 3, line 82: I assume it shall be Fig. 1A and not S1A?

- line 102: energy-independent instead of energy-independently
- line 105: ...binding to pockets on its NTD...? ATP's pockets? Please rephrase.
- page 5, from line 122: please split this sentence into at least 2!
- page 9, line 228: residues not residue
- page 10, line 254: Very recently, we have shown
- page 12, line 308: remove word "became"
- page 21, from line 506: The sentence starting with "Therefore" is weird and needs attention on language and content.
- page 25, line 593: were expressed
- Are the blue boxes in Figures 4, 6, 7, 8 and 9 necessary?
- Fig. S1: This should be RNA sequences!
- Suppl. Table: I assume Kd is meant as constant (if not rate is meant, which would be kd)!

Reviewer #3 (Remarks to the Author):

The manuscript titled "A unified mechanism for LLPS of SARS-CoV-2 N protein as specifically modulated by ATP and nucleic acid" by Dang, Li, and Song described the liquid-liquid phase separation behaviour of protein of SARS-COV2 (nucleocapsid (N) protein). In this work, the authors dissected N protein into differential combinations of domains that could produce well-dispersed NMR spectra and then followed the LLPS by DIC and NMR. The N-terminal construct (N1-249) produced high-quality NMR spectra and was also phase separated. The LLPS was modulated by ATP and Nucleic acid.

Authors demonstrated that ATP and nucleic acid interplay in modulating LLPS happens via specific competitions for the same binding site. Finally, the authors proposed that the mechanism of LLPS of N protein can be targetable by small molecules. Though the work appears to be reasonably well-conducted but lack several clarities. Below is the summery for the shortcomings of this work:

1. General observation: Authors have picked two trendy terms to combine in this work --- LLPS and SARS-COV2. Is phase separation established in viral infection or to be more specific, influenza infection?
2. LLPS of IDR by nucleic acid has already been shown for FUS protein. Have authors done low complexity domain of analysis for all proteins of SARS-COV2 to derive the logic behind N-protein phase separation behaviour.
3. Authors showed that the binding of ATP is in ~ 3 mM range, then how could it replace Nucleic acids, which tightly bind to IDR?
4. How authors can claim that their study will be useful in designing a drug that can handle various SARS-COV2 variants. Can then establish even for one variant?
5. In the NMR spectra, how much signal comes from monomer and how much from the higher molecular weight species. Can they propose a quantitative estimate?
6. Can the authors show that supplementation of ATP can inhibit invasion and propagation of SARS-COV2?

In summary, the work presented here has been done carefully but lacks a better and bigger perspective.

Dear Reviewers

Thank you very much for your kind time and comments at this challenging moment. I would apologize for taking so long to revise the manuscript, because it was also an extremely challenging time for my group.

In the past, we have devoted great efforts to conduct all experiments the reviewers kindly suggested. Unfortunately one student got infected despite being triple-vaccinated and took a while to recover. Another student got an abnormal heart condition after vaccination and had to take medical leave.

Now we have completed experiments and conducted thorough revisions by adding new experiments, references and discussion to address all the comments. The detailed responses are documented below.

With kindest regards!

Reviewer #1

1) Abstract: “ATP is not only emerging as a cellular factor controlling the host-SARS-CoV-2 interaction, but also provides a lead for developing anti-SARS-CoV-2 drugs efficient for different variants of SARS-CoV-2.” This comment is way too forward compared to the extent of the work. This work is mainly based on NMR and DIC. Whether NMR can efficiently study all the interactions and alterations, DIC is not proven the most efficient method to characterize a LLPS. Authors’ approach to quantitatively detect the residue-specific interaction through NMR is exciting, but it cannot be established as lead for drug at this time. For this, it was needed some amount of cellular work to prove the effect of ATP in condensate formation N protein with at least an over-expression construct. Also, there should have more microscopic and other experiments in order to prove LLPS formation and nature of liquid droplets in presence and absence of nucleic acid and ATP.

Response: thanks so much for the comment. We agree and have removed this statement from the abstract.

2) Introduction: Line 102, “energy-independently” should be written as “energy-independent”.

Response: thanks for pointing out the mistake. We have corrected it.

3) Line 105, “specifically” should be written as “specific”.

Response: thanks and we have corrected it.

4) Line 114, Here it is represented as “SFig 1A” where in line 132, it is as “Fig S1B”. Please make the pattern same throughout the whole manuscript.

Response: thanks and we have corrected it.

5) Line 122-127, “Despite the criticality of understanding LLPS of N protein for developing therapeutic strategies/molecules to fight the pandemic, the high-resolution mechanisms for LLPS of N protein still remain unknown most likely due to the challenge in characterizing the association-prone N protein by the high-resolution biophysical methods particularly by NMR spectroscopy, the only one available so far to experimentally obtain the residue-specific knowledge of LLPS (12,21-28).” This sentence is way too long. Please break it down and make it easy for the understanding of readers.

Response: thanks for the comment. We have revised the sentence.

6) Results: Line 182, Fig 1C, 1D X and Y- scale font sizes are very different, hard to understand. Superimposable images are hard to understand, please use arrows to indicate specific changes in specific residues. Author can use specific color arrows for specific residue changes.

Response: thanks for the comment. We have revised the figure.

7) In Figure 1C, The comparison of the 1D spectra of different variants of N protein is missing a few important points. Specifically,

a) Chemical shift referencing with standard sample is recommended for accurate measurement of the methyl chemical shifts.

b) The whole region of the sidechain is missing from the comparison.

c) 1D spectra of the buffers must be added as a control spectra.

Response: thanks so much for the comments. a) We have added it in the Method section. b) The whole side-chain region is not comparable because the N protein fragments containing IDRs have very huge peaks from methyl groups of the unstructured residues which would make the very up-field peaks from the folded domains too small to be seen. c) In our study, phosphate buffer was used and therefore the buffer has no proton NMR signal in any 1D spectra.

8) Line 184-185, “NTD are highly similar in both isolated NTD and N (1-249)”- Authors should add a comparison of 1H-13C HSQC spectra, importantly around the methyl proton region.

Response: thanks so much for the comment. We have attempted to collect 1H-13C HSQC spectra for N protein fragments. However, there are two intrinsic problems with the 1H-13C HSQC spectra for fragments containing IDRs. 1) The expression levels of some fragments were very low in the medium used for double-labelling the proteins. 2) The protein fragments containing IDRs have strong 1H-13C HSQC peaks from methyl groups of the unstructured residues which would make the up-field peaks from the folded domains too weak to be detected.

On the other hand, the sensitivity of proton 1D spectra is much higher for detecting the very up-field peaks which is sufficient here to indicate the existence of the folded NTD and CTD.

9) Line 190, “CTD is also similarly folded in the isolated domain” There is no significant experimental evidence found for this statement. Both the 1D and 2D spectra are significantly different in N (175-419) and CTD (247-364). It looks like N (175-419) is unstructured compared to only CTD (247-364) construct. Also, as the author shows the 1D spectra of the CTD constructs are similar (Fig S3A), it is recommended to show a 1H-13C HSQC comparison.

Response: thanks so much for the comment. As justified above, the 1H-13C HSQC spectra have the intrinsic problems and are thus not informative for IDR-rich proteins. On the other hand, to address the comment we have collected new 1H-15N HSQC spectra with 512 scans whose HSQC peaks of the folded CTD are mostly detectable and superimposable to those of the isolated CTD,

We have added the spectra with assignments in Fig. S3, which now provides the strongest evidence that CTD in different fragments is similarly folded.

10) Line 194, “Well dispersed peaks”, Can the authors assign those well-dispersed peaks to know the residues?

Response: thanks so much for the comment. Yes, we have collected new 1H-15N HSQC spectra with 512 scans whose HSQC peaks of CTD are detectable and superimposable to those of the isolated CTD, We have thus added the spectra with assignments in Fig. S3.

11) Line 218, Authors have justified their use of high ATP concentration as it is physiologically relevant. But they have use protein N concentration around 100 μ M or so. Please shed light on the concentration of N protein physiologically and talk a little bit about the stoichiometry of protein and ATP in their experiments.

Response: thanks so much for the comment. The concentrations of N protein appears to be variable at different stages of infection. In fact we have discussed this in our previous paper in BBRC and explored its relevance to the roles of ATP. As the current manuscript is focused

on the mechanism of LLPS, we did not include a detailed discussion. In particular, the second reviewer kindly requested to significantly shorten the manuscript.

“The result not only establishes the SARS-CoV-2 RBD to be the first viral domain capable of binding ATP at biologically-relevant concentrations (\sim mM), but also suggests that RBD may have an pivotal role in specifically regulating the uncoating, localizing and packing of the genomic RNA by the N protein. As illustrated in Fig. 4E, immediately after the infection, the SARS-CoV-2 will release its genomic RNA-N-protein condensate into the infected cell, which is tightly packed into the gel-like state [5]. With consideration that at this stage, one infected cell may only have one to several copies of the condensate, the ratios between ATP and N protein/genomic RNA are very high. Consequently ATP acts to facilitate the condensate to be uncoated, such as to transform the gel-like condensate into more dynamic liquid droplets or even homogenous solution. Furthermore, once new copies of viral RNA polymerase and N-protein are synthesized by the host cell machinery, the ratios will reduce, and ATP may enhance LLPS of the mixture of the viral genomic RNA and N proteins as well as the host cell replicases to form replicase-transcriptase complexes. Finally, after all components needed for the assembly of new virions are synthesized by the infected cell, the ratios between ATP and N protein/genomic RNA will be further reduced and therefore a large population of the ATP-RBD complex will become dissociated. As such, the ATP-unbound RBD of the SARS-CoV-2 N protein become available for binding the specific sites of the genomic RNA to initiate the packing process, which might be even enhanced by ATP at low molar ratios.”.

12) Line 221, “HSQC peaks underwent significant shift as illustrated by I of Fig. 2”, Superimposition of HSQC spectra of 6 different concentrations of ATP will be useful to track ATP dependent shift of the amide peaks.

Response: thanks so much for the comment. We have prepared a new figure with assignment and different ATP concentrations (Fig. S4) as kindly suggested.

13) In III of Fig 2, The peaks in the IDRs become resolved upon 10 mM ATP binding. This signifies significant structural change in IDRs. The authors should add more insights on this.

Response: thanks so much for the comment. We have prepared Fig. S4 as the reviewer kindly suggested. In the spectra, ATP only triggered shifts of some IDR 1H-15N HSQC peaks upon binding but failed to change their overall spectral dispersion. Therefore, no significant structure change is expected to occur.

14) Line 227-235, Authors state that residues with significant shift have $CSD > 0.072$. However, in their dissociation constant calculation they select only 10 NTD residues. Surprisingly, lots of residues in NTD with $CSD > 0.072$ were not included in the calculation. How does they affect the dissociation constant?

Response: thanks so much for the comment. We would apologize for confusion. In fact, as shown in Table S1, we included all 10 residues for calculation in the context of N (1-249). Some large CSDs are in fact from residues of the isolated NTD. So to clarify the confusion, we have labelled all 10 residues of N (1-249) in Fig. 2A.

15) In the legend of “II of Figure 2”, isolated NTD (Green) is not mentioned.

Response: thanks so much for the comment. We have prepared a new figure with assignment and different ATP concentrations (Fig. S4).

16) Line 232, it will be good to shed the light on the reason of exclusion of Ala90 in this work.

Response: thanks so much for the comment. We would apologize for the confusion. This is typo and it should be Leu56 but not Ala90. We have corrected it. In fact, as shown in Table S1, we included all 10 residues including A90 for calculation.

17) There was no clear difference in the 1D spectra of free and ATP-bound states in Fig. S4A.

Response: thanks so much for the comment. As ATP has a very low binding affinity (K_d of mM), Significant shifts were observed only on HSQC peaks while only very slight shift was observed on one up-field peak.

18) Authors' explanation on the involvement of Arg residues are interesting, but there are many reports on binding to Arginine leads to folding of IDR, which may lead to aggregation/oligomerization. Please shed light on this fact how oligomerization and LLPS are controlled.

Response: thanks so much for the comment. As in the present study, the binding to Arg only trigger some shifts of HSQC peaks but not an overall increase in the spectral dispersion of IDRs. This suggests that no large structure change occurs in our system.

With regard to aggregation, the contribution of Arg is extremely complex and context-dependent. For example, our previous studies on TDP-43 PLD showed that either removing or mutating Arg to Ala resulted in protein aggregation (Dang, Lim, Kang, & Song, 2021). Furthermore, ATP-binding to them in fact acts to significantly inhibit aggregation.

In the fibril structure of TDP-43 PLD, the side chains of the Arg in neighbor layers are in close spatial position. Therefore, the positive charges of Arg side chains are expected to repel each other, thus acting to inhibit aggregation/fibrillation. On the other hand, the ATP-binding introduced conformation-specific negative charges of ATP, which also produced strong repulsive electrostatic interactions to inhibit aggregation.

19) Line 283, Continuing from the previous point, turbidity is also an indication of aggregation. Can author provide some references on turbidity as a measurement of LLPS?

Response: thanks so much for the comment. The turbidity assay is commonly recommended to assess protein LLPS (Huang et al., 2021; Wang, Zhang, & Zhang, 2019). To avoid the common pitfalls of this assay exceeding the linear detection range, we used the protein at a low concentration to get rid of aggregation and combined it with the direct DIC imaging to improve the accuracy of the readouts.

20) Line 283, DIC imaging is not the best way to confirm LLPS. Microscopy with labeled protein can confirm the phase separation. Some more works are suggested in order to characterize LLPS. Some videos on fusion of the liquid droplets are handy along with FRAP to show the nature of LLPS.

Response: thanks so much for the comment. In fact, previously we have conducted the fluorescent-labelling of a couple of proteins for LLPS studies. Unfortunately we found that the introduction of fluorescent agents all with aromatic rings will fundamentally altered the properties/mechanisms of LLPS.

Most strikingly, the FUS C-terminal RGG domain which contains Arg residues but no aromatic residues is unable to phase separation by itself. However, once we conducted the fluorescent labelling by introducing chemicals with aromatic rings (all chemicals with visible

fluorescence contain aromatic rings), the RGG domain suddenly became able to phase separate by itself. Now with results from many groups including us it is clear this is because for the IDRs, the strongest driving forces are π - π and π -cation interactions between Arg side chain and aromatic rings regardless from aromatic amino acids, DNA/RNA or fluorescent chemicals.

Therefore, for our high-resolution mechanistic study, fluorescent labelling will unavoidably lead to establishing non-native interactions and consequently generating artifacts.

21) A major question is, if using high concentration of nucleic acid (S2m) itself is capable of dissolve LLPS, then what is the need of ATP here to replace the interactions. In this case, adding more nucleic acid to the nucleic binding sites would help.

Response: thanks so much for the comment. Yes, in our biophysical experiments, we can dissolve LLPS by adding more nucleic acids. However, in cells, the concentrations of nucleic acids are tightly regulated and also much lower than that of ATP. So ATP represent a key cellular factor to interplay with nucleic acids in modulating LLPS. In fact, we have discussed this issue in our previous paper in BBRC and explored its relevance to the roles of ATP in virus-host interactions. As the current manuscript is focused on the mechanism of LLPS, we did not include such a discussion.

“The result not only establishes the SARS-CoV-2 RBD to be the first viral domain capable of binding ATP at biologically-relevant concentrations (\sim mM), but also suggests that RBD may have an pivotal role in specifically regulating the uncoating, localizing and packing of the genomic RNA by the N protein. As illustrated in Fig. 4E, immediately after the infection, the SARS-CoV-2 will release its genomic RNA-N-protein condensate into the infected cell, which is tightly packed into the gel-like state [5]. With consideration that at this stage, one infected cell may only have one to several copies of the condensate, the ratios between ATP and N protein/genomic RNA are very high. Consequently ATP acts to facilitate the condensate to be uncoated, such as to transform the gel-like condensate into more dynamic liquid droplets or even homogenous solution. Furthermore, once new copies of viral RNA polymerase and N-protein are synthesized by the host cell machinery, the ratios will reduce, and ATP may enhance LLPS of the mixture of the viral genomic RNA and N proteins as well as the host cell replicases to form replicase-transcriptase complexes. Finally, after all components needed for the assembly of new virions are synthesized by the infected cell, the ratios between ATP and N protein/genomic RNA will be further reduced and therefore a large population of the ATP-RBD complex will become dissociated. As such, the ATP-unbound RBD of the SARS-CoV-2 N protein become available for binding the specific sites of the genomic RNA to initiate the packing process, which might be even enhanced by ATP at low molar ratios.”

22) Line 341-343, “provoked μ s-ms dynamics”, “IDR residues have rather independent dynamic behaviors” Line 345-346, “the large dynamically-crosslinked complexes”- Can author provide significant references on these comments?

Response: thanks so much for the comment. “provoked μ s-ms dynamics”: Dinesh et al. reported that the NTD of SARS-CoV-2 N protein binds the RNA (5'-UCUCUAAACG-3') with a K_d value of μ M (8.3 ± 0.8) (Dinesh et al., 2020). This is consistent with our results that S2m binding triggers linewidth broadening as the binding affinity is within intermediate exchange regime.

“IDR residues have rather independent dynamic behaviors”: In contrast to globular proteins with hydrophobic cores which are buried and inaccessible, IDRs exist as diverse and dynamic ensembles of interchanging structural conformations. Please kindly refer to the literature

(Davey, 2019). “the large dynamically-crosslinked complexes”: The droplets formed through LLPS are large, dynamically cross-linked complexes; for this point, please kindly refer to the paper (Alberti, Gladfelter, & Mittag, 2019).

23) In Figure 6C, it looks like their representation after dissociation of LLPS does not support their data. Their data shows, after dissociation there is only existence of binding to IDRs, which is not specifically shown in the figure.

Response: thanks so much. In fact, the results suggest that even after the dissolution of LLPS by excessive binding with nucleic acids, the NTD, and Arg/Lys residues over IDRs still dynamically bind with nucleic acids. Therefore, the intensities of the HSQC peaks for these residues were not restored.

24) Line 377, Both 1:500 and 1:750 is very high. Please shed light on the desired physiological stoichiometry.

Response: thanks so much for the comment. As provided above, we have discussed the interplay of ATP and nucleic acids in our previous paper in BBRC and explored its relevance to the roles of ATP. As the current manuscript is focused on the mechanism of LLPS, we did not include such a discussion.

25) Line 403, What does it mean by “significant increase of solution viscosity”? How is it measured? Is there any reference to this point?

Response: thanks so much for the comment. It has been previously shown that ATP at high concentration can form cluster with itself as well as with water, thus increase the solution viscosity. To address this, we have included a reference here.

26) Line 413, “ATP at 20 mM”, The NMR measurements was ~2.0 mM. Is it a typo?

Response: thanks so much for the comment. It is 20 mM. Here we aimed to interrogate the effect of ATP on the displacement of the binding of S2m from the protein. The NMR titration points for ATP included 5, 10, 15, and 20 mM, and we found even with the addition of ATP up to 20 mM, it could not completely restore the disappeared peaks induced by S2m.

27) Line 646, Is it CTD or NTD?

Response: thanks so much for the comment. It is CTD.

Reviewer #2

1) Assuming ATP to have a strong physiological role in SARS-CoV-2 LLPS, what about its cofactor Mg²⁺? Could authors provide evidence for effects of magnesium concentration taking into account typical cellular concentrations? Along with that, authors will need to look at the effects of ATP (and Mg) on the secondary structure of S2m as it might be an indirect effect (see later comment on RNAs) for the LLPS phenotype.

Response: thanks so much for the comment. We have collected the spectra as the reviewer kindly suggested and summarized in Fig. S5. No interaction was observed for MgCl₂ on N (1-249) (A) and CTD (B) as well as ATP on S2m (C).

2) Seeing the high concentrations necessary to titrate N protein, I wonder about the stability of pH in NMR samples (knowing about N's susceptibility to changes in pH). Has pH been monitored and can potential effects on CSDs and/or LLPS be separated from ATP?

Response: thanks so much for the comment. Please kindly refer to the Methods and Materials, in all our NMR titration experiments, ATP, nucleic acids stocks, and proteins were prepared in the same buffer (10 mM NaPi, 150 mM NaCl, pH 6.8) close to physiological conditions. In turbidity assay and DIC imaging, ATP, nucleic acids stocks, and proteins were prepared in droplets buffer (25 mM HEPES, pH 7.5, 70 mM KCl) (Carlson et al., 2020). As a consequence, the titration studies using ATP or nucleic acids are expected to lead to definitely no pH changes.

3) Have the authors considered to include a construct of NTD-IDR2-CTD? According to very recent work (Bessa et al., Sci Adv., 2021) this should be NMR-suitable and would combine the features of specific binding by the NTD, dimerization by the CTD and RNA-binding through IDR2 together with both folded domains binding ATP? Please comment at least!

Response: thanks so much for the suggestion. We have devoted great efforts to cloning and expressing this construct as the reviewer kindly suggested. However, unfortunately this construct appeared to be extremely prone to aggregation and got completely precipitated upon adding ATP and S2m even at low concentrations.

4) In order to more precisely discuss the role of host ATP for the viral life cycle: I assume there meanwhile is sufficient literature available on proteome changes after SARS-CoV-2 infection, which might give hints on differential levels of ATP synthetases/ATPases? Could the authors comment on that?

Response: thanks so much for the comment. According to the literature, SARS-CoV-2 proteins interact with molecular complexes involved in intracellular trafficking and transport as well as cellular metabolism (Stukalov et al., 2021). In concert with these findings, SARS-CoV-2 infection has been reported to induce a shift of the host cell metabolism towards enhanced glycolytic activity, with an increase in hexokinases 1 and 2 but a decrease in fructose-1, 6-bisphosphatase 1 and 2 (Bojkova et al., 2021). This hyperglycolysis and reprogramming in metabolism not only promote the production of NADPH and ROS in the pentose phosphate pathway, but also further exacerbates inflammation (Santos, Póvoa, Paixão, Mendonça, & Tabora-Barata, 2021). All of these processes are implicated in the production and use of ATP. However, the quantitative and dynamic analysis of ATP

biosynthesis and consumption with COVID-19 progression still remain extremely challenging to be accomplished.

Therefore, after a careful consideration, we decided not to discuss these much beyond our biophysical study. In particular, the reviewer kindly requested to shorten the current manuscript.

5) What is the rationale of using S2m here, it is not the bona fide target of N in SARS-CoV-2 (or is there a reference missing? the reference to the own previous paper does not really help)!? Would one not expect to see even more drastic effects with larger RNAs? Please compare findings (in DIC) using a short and long (ideally more/less structured) RNA. Would one need a much higher excess of ATP to dissolve LLPS with large RNAs? I do not demand a full experimental setup considering all parameters like N oligomeric states and the role of RNA-structural content. But it appears too selective to only look at one RNA only. I also did not understand, if CTD and NTD on their own bind to S2m? What are the affinities then?

Response: thanks so much for the comment. Previously in our other papers, we have tested nucleic acids of different sequences and variable lengths. However, most reviewers commented that for our biophysical studies, we should stick to using one biologically-relevant sequence in order to enhance the biological relevance of our results and in the meanwhile to reduce the unnecessary efforts and fund cost. Indeed, introduction of one extra sequence need to do a full set of experiments which demands huge efforts, time and fund.

Therefore, here we selected the 32-mer stem-loop II motif (S2m) within the genome 3-UTR because it is a highly conserved sequence among CoVs and confers selective advantages. It has been corroborated to interact with SARS-CoV N protein and extensively used in previous N protein-RNA/DNA binding studies (Chen et al., 2007).

6) Where was S2m taken from or how was it produced? This is not given in the Methods section.

Response: thanks so much for the comment. The sequence information of S2m is from the literature (Chen et al., 2007). It was synthesized in vitro by Integrated DNA Technologies, Inc. Here we used its DNA mimic to get rid of RNase contamination and degradation.

7) I am puzzled by the usage of the term specificity in the manuscript. It might not be correct or actually needed in all places where used. E.g. what is the “specific” binding of ATP to NTD and IDRs? Isn’t it almost all Args in the end? This term “specific” needs to be discussed. (see also page 24, line 564) In particular for the NTD, how can it be specific if only the hinge region of the hairpin is affected and not the hairpin residues themselves (which have been shown to interact with RNA “specifically”)? Is this meant with specific? And if ATP is not bound by all positive residues, how would that be explained within IDRs? Please discuss! Along the same route: page 16, line 406: why does S2m “specifically” interact with 1-249? Compared to other RNAs or proteins? What is meant here, since N will bind lots of other RNAs from SARS-CoV-2 as well!

Response: thanks so much for the comment. The term specificity here indicates that the interaction between ATP/S2m and the protein is residue-specific, which means that ATP can selectively bind certain residues over others. This is in contrast to the non-specific types of interactions, such as hydrophobic interactions. Our NMR HSQC experiments clearly suggest this point, as the chemical shift changes upon adding ATP/S2m are both residue dependent (NTD and Arg/Lys residues over IDRs) and multidirectional within the ¹H, ¹⁵N plane, which is typically expected for specific interactions. In contrast, nonspecific interactions or solvent

effects often cause unidirectional chemical shift changes with reduced residue-dependence. For more information, you may refer to the literature (Boulton et al., 2019).

8) On affinities: How is the higher affinity for 1-249 for ATP explained when looking at the NTD peaks? Should those not be in competition with binding sites in IDRs? (page 9, bottom statement). This does not completely convince me in its current description. Have different oligomer states between constructs (also important for interpreting LLPS) been taken into consideration? In fact, I do not agree that NTD affinity for ATP is higher in N1-249, looking at the provided spectra. Maybe the authors could corroborate their calculations and statements with corresponding raw data (e.g., seeing the two peaks between 9.35 and 9.5 / 121 and 123).

Response: thanks so much for the comment. First, there is only a slight affinity difference and therefore in the revised manuscript, we removed the emphasis on this. Second, In contrast to the isolated NTD with wildly fluctuating N-terminal and C-terminal residues, the movements of the corresponding residues in N (1-249) are restricted by the NIDR and central IDR. This may give rise to higher ATP-binding affinities for these residues. On the other hand, the dynamics of the isolated NTD and the NTD in the context of N (1-249) should be different; similar situations have been reported in the tandem RRM domains of TDP-43 and hnRNPA1. Such differences in dynamics may also result in variations in binding affinities.

9) Regarding NMR titrations: why would some peaks not reappear even at high excess of RNA? (page 13, top) Please discuss in a bit more detail (exchange?)! I also suggest to include an intensity plot of free vs. recovered spectra after LLPS instead of just mentioning it along with full spectral views? It should be shown as those data are available.

Response: thanks so much for the comment. Please kindly refer to page 14, line 346-351. The higher-order oligomeric protein structures were disrupted after adding excessive S2m to dissolve the N protein LLPS. This dissolution relieved protein-protein association; therefore, the peak intensities of most residues over IDRs were restored.

However, the NTD and Arg/Lys residues over IDRs were still bound with S2m. Given that the binding affinities of S2m to the NTD and Arg/Lys residues are within the micromolar range, this induces intermediate exchange and therefore triggers the broadening of these peaks.

In fact, what Fig. 4A (original Fig. 6A) presents are the intensity ratios of HSQC peaks of bound/recovered versus free states.

10) On Figures: Figures should be designed/labeled (inset text) to express what is claimed. For some of them, I suggest to provide additional zoom-ins rather than sole full-spectral views. This includes the Kd determination: Exemplary shifting residues should be shown indicating the data quality (seeing the minor CSPs present at maximal ATP concentration). In its present form the large number of HSQCs hardly makes a link to the derived data.

Response: thanks so much for the comment. We have extensively revised figures.

11) I strongly suggest shortening of the manuscript. It appears very redundant in information throughout the intro, results and discussion and readability will improve by removing e.g. too much interpretation in the intro. Also the language style seems to shift in between (which is ok for me), and there are many incomplete sentences (see minor issues for examples).

Response: thanks so much for the comment. We have extensively shortened the manuscript.

12) There is insufficient handling of available literature and referencing per se. E.g., the second paragraph of the introduction cites 15 references in a block, while e.g. line 80 misses a good point to more specifically put a reference. Please be more precise here in what references refer to what finding! The same for page 3, line 88: I do not agree that those citations claim LLPS to be a key mechanism for the diverse viral functions, although it might be a strong indication. Also: page 4, line 92: please add a reference!

Response: thanks so much for the comment. Please kindly find the revised references below. (Line 75-92) “N protein is the only structural protein which not only functions to package gRNA, but is also responsible for suppressing the immune system and manipulating the cellular machineries of the host cell to enhance the viral infection and replication (Masters, 2019; McBride, van Zyl, & Fielding, 2014; Shah, Firmal, Alam, Ganguly, & Chattopadhyay, 2020). For example, N protein has been currently identified to play critical roles in hijacking the host cell machineries for RNA replication and transcription (Z. Bai, Y. Cao, W. Liu, & J. Li, 2021; Cascarina & Ross, 2020), nucleocapsid assembly (Ye, West, Silletti, & Corbett, 2020), virion assembly and virion package (Chang, Hou, Chang, Hsiao, & Huang, 2014; Jack et al., 2021). Furthermore, N protein provides the connection between viral E, M proteins and gRNA within virion (Siu et al., 2008). In particular, it has high immunogenicity (4) and a low rate of mutation (Fig. S1A), with 91% identity to that of SARS-CoV-1, which is much more conserved than the S protein. Consequently, N protein represents a key candidate for future drug and vaccine development (Zhihua Bai, Ying Cao, Wenjun Liu, & Jing Li, 2021; Dutta, Mazumdar, & Gordy, 2020).

Most strikingly, liquid-liquid phase separation (LLPS), the emerging principle for commonly organizing the membrane-less organelles (MLOs) critical for cellular physiology and pathology (17-19), has been very recently identified as the key mechanism underlying the diverse functions of SARS-CoV-2 N protein. For example, SARS-CoV-2 N protein has been previously shown to phase separate upon introducing various nucleic acids of specific and non-specific sequences (Caruso et al., 2022; Iserman et al., 2020).”

We have added all these references in the revised manuscript.

Minor points are:

- I find the title misleading. Unified to or with what?
- inconsistency in labeling in Fig. 1: NTD 40-180 vs. legend NTD 44-180
- page 3, line 67: numbers of infections and deaths based from SARS-CoV-2 should be referenced with a particular day looked at
- page 3, line 82: I assume it shall be Fig. 1A and not S1A?
- line 102: energy-independent instead of energy-independently
- line 105: ...binding to pockets on its NTD...? ATP's pockets? Please rephrase.
- page 5, from line 122: please split this sentence into at least 2!
- page 9, line 228: residues not residue
- page 10, line 254: Very recently, we have shown
- page 12, line 308: remove word “became”
- page 21, from line 506: The sentence starting with “Therefore” is weird and needs attention on language and content.
- page 25, line 593: were expressed
- Are the blue boxes in Figures 4, 6, 7, 8 and 9 necessary?

- Fig. S1: This should be RNA sequences!
 - Suppl. Table: I assume K_d is meant as constant (if not rate is meant, which would be k_d)!
- Response:** thanks so much for the comment. We have changed the title to be “**Mechanism for ATP and nucleic acid to interplay in modulating LLPS of SARS-CoV-2 N protein**”. We also corrected them.

Reviewer #3

1. General observation: Authors have picked two trendy terms to combine in this work -- LLPS and SARS-COV2. Is phase separation established in viral infection or to be more specific, influenza infection?

Response: thanks so much for the comment. There is accumulating evidence that condensates/compartments are widely involved in viral infections, not only in influenza, but also in others, such as vesicular stomatitis virus, Borna disease virus, and rabies virus (Heinrich, Maliga, Stein, Hyman, & Whelan, 2018; Hirai, Tomonaga, & Horie, 2021; Nikolic et al., 2017). The condensates/compartments are formed through LLPS, also known as viral factories, viroplasm, or viral replication centers, which function in viral genome replication, gene expression and antiviral immune response to maximize the ability to propagate in hosts. For more information, you may kindly refer to the review paper (Alberti & Dormann, 2019; Brocca, Grandori, Longhi, & Uversky, 2020).

We have added all these references in the revised manuscript.

2. LLPS of IDR by nucleic acid has already been shown for FUS protein. Have authors done low complexity domain of analysis for all proteins of SARS-COV2 to derive the logic behind N-protein phase separation behaviour.

Response: thanks so much for the comment. We have used PlaToLoCo: the first web meta-server for visualization and annotation of low complexity regions in proteins

3. Authors showed that the binding of ATP is in ~ 3 mM range, then how could it replace Nucleic acids, which tightly bind to IDR?

Response: thanks so much for the comment. Although ATP has a binding affinity in the millimolar range compared to nucleic acids with binding affinities in the micromolar range, the binding interactions between nucleic acids and the N protein are still dynamic. Furthermore, the concentrations of nucleic acids are tightly regulated and much lower than ATP. Therefore, it is expected that ATP at high concentrations can compete and replace the binding of nucleic acids.

4. How authors can claim that their study will be useful in designing a drug that can handle various SARS-COV2 variants. Can then establish even for one variant?

Response: thanks so much for the comment. According to our results, ATP binding pocket is constituted by the residues that are highly conserved in SARS-CoV-2 and emerged variants (see in .ppt page 11). This suggests that the mechanism by which ATP modulates the N protein LLPS should be universal among SARS-CoV-2 and variants.

As such, anti-SARS-CoV-2 molecules might be designed to occupy ATP binding pocket but with much higher affinity, which thus inhibit the essential functions of N protein including to block its binding with nucleic acids and to disrupt LLPS.

5. In the NMR spectra, how much signal comes from monomer and how much from the higher molecular weight species. Can they propose a quantitative estimate?

Response: thanks so much for the comment. Previous studies indicated that CTD is a very stable dimer. In other words, in NMR chemical shift time scale, it exists only in the dimeric state and consequently all NMR signals for CTD are from the dimeric form.

6. Can the authors show that supplementation of ATP can inhibit invasion and propagation of SARS-COV2?

Response: thanks so much for the comment. First, we are a biophysical group who does not have the permission and setting to do SARS-CoV-2 in vivo experiments. Second, I have discussed with some colleagues who are hardcore virologists but they thought it is extremely challenging to do this experiment because the change of ATP concentration might affect millions biological processes in cells. Third, this experiment is much beyond the scope of our current manuscript.

References

Alberti, S., & Dormann, D. (2019). Liquid–liquid phase separation in disease. *Annual review of genetics*, 53, 171-194.

Bai, Z., Cao, Y., Liu, W., & Li, J. (2021). The SARS-CoV-2 Nucleocapsid Protein and Its Role in Viral Structure, Biological Functions, and a Potential Target for Drug or Vaccine Mitigation. *Viruses*, 13(6). doi:10.3390/v13061115

Bojkova, D., Costa, R., Reus, P., Bechtel, M., Jaboreck, M.-C., Olmer, R., . . . Cinatl, J. (2021). Targeting the pentose phosphate pathway for SARS-CoV-2 therapy. *Metabolites*, 11(10), 699.

Boulton, S., Selvaratnam, R., Ahmed, R., Van, K., Cheng, X., & Melacini, G. (2019). Mechanisms of specific versus nonspecific interactions of aggregation-prone inhibitors and attenuators. *Journal of medicinal chemistry*, 62(10), 5063-5079.

Brocca, S., Grandori, R., Longhi, S., & Uversky, V. (2020). Liquid–liquid phase separation by intrinsically disordered protein regions of viruses: Roles in viral life cycle and control of virus–host interactions. *International Journal of Molecular Sciences*, 21(23), 9045.

Cascarina, S. M., & Ross, E. D. (2020). A proposed role for the SARS-CoV-2 nucleocapsid protein in the formation and regulation of biomolecular condensates. *The FASEB Journal*, 34(8), 9832-9842. doi:https://doi.org/10.1096/fj.202001351

Davey, N. E. (2019). The functional importance of structure in unstructured protein regions. *Current opinion in structural biology*, 56, 155-163.

Dinesh, D. C., Chalupska, D., Silhan, J., Koutna, E., Nencka, R., Veverka, V., & Boura, E. (2020). Structural basis of RNA recognition by the SARS-CoV-2 nucleocapsid phosphoprotein. *PLoS pathogens*, 16(12), e1009100.

Heinrich, B. S., Maliga, Z., Stein, D. A., Hyman, A. A., & Whelan, S. P. (2018). Phase transitions drive the formation of vesicular stomatitis virus replication compartments. *Mbio*, 9(5), e02290-02217.

Hirai, Y., Tomonaga, K., & Horie, M. (2021). Borna disease virus phosphoprotein triggers the organization of viral inclusion bodies by liquid-liquid phase separation. *International Journal of Biological Macromolecules*, 192, 55-63.

Huang, Y., Bai, Y., Jin, W., Shen, D., Lyu, H., Zeng, L., . . . Liu, Y. (2021). Common Pitfalls and Recommendations for Using a Turbidity Assay to Study Protein Phase Separation. *Biochemistry*, 60(32), 2447-2456.

- Masters, P. S. (2019). Coronavirus genomic RNA packaging. *Virology*, 537, 198-207. doi:<https://doi.org/10.1016/j.virol.2019.08.031>
- McBride, R., van Zyl, M., & Fielding, B. C. (2014). The coronavirus nucleocapsid is a multifunctional protein. *Viruses*, 6(8), 2991-3018. doi:10.3390/v6082991
- Nikolic, J., Le Bars, R., Lama, Z., Scrima, N., Lagaudrière-Gesbert, C., Gaudin, Y., & Blondel, D. (2017). Negri bodies are viral factories with properties of liquid organelles. *Nature Communications*, 8(1), 1-13.
- Santos, A. F., Póvoa, P., Paixão, P., Mendonça, A., & Taborda-Barata, L. (2021). Changes in glycolytic pathway in SARS-COV 2 infection and their importance in understanding the severity of COVID-19. *Frontiers in chemistry*, 9.
- Shah, V. K., Firmal, P., Alam, A., Ganguly, D., & Chattopadhyay, S. (2020). Overview of Immune Response During SARS-CoV-2 Infection: Lessons From the Past. *Frontiers in Immunology*, 11. doi:10.3389/fimmu.2020.01949
- Siu, Y., Teoh, K., Lo, J., Chan, C., Kien, F., Escriou, N., . . . Peiris, J. (2008). The M, E, and N structural proteins of the severe acute respiratory syndrome coronavirus are required for efficient assembly, trafficking, and release of virus-like particles. *Journal of virology*, 82(22), 11318-11330.
- Stukalov, A., Girault, V., Grass, V., Karayel, O., Bergant, V., Urban, C., . . . Wang, A. (2021). Multilevel proteomics reveals host perturbations by SARS-CoV-2 and SARS-CoV. *Nature*, 594(7862), 246-252.
- Ye, Q., West, A. M., Silletti, S., & Corbett, K. D. (2020). Architecture and self-assembly of the SARS-CoV-2 nucleocapsid protein. *Protein Science*, 29(9), 1890-1901.

Reviewers' comments:

Reviewer #1 (Remarks to the Author):

Authors has mostly addressed the concerns. Although, there are some loopholes of the studies and they could not address those yet. Some of the points are-

1) Although, they have not been able to show some of the comparison ^1H - ^{13}C spectra, they have been able to add a new ^1H - ^{15}N spectra. But, this makes it a little bit more confusing to compare and understand the actual interactions.

2) On the concern about stoichiometry, authors gave a previous reference which does not answer the concerns. It's all about a model which can replicate the cellular/physiological concentration or ratio that works between each other. This is still a loophole. Some cellular work could have answered the combinational effect.

3) Author mentioned the corresponding references (which were requested to be added) in the rebuttal letter. Authors should make sure that they have also added those in the manuscript.

4) On the question of how did they measure viscosity change, the response was not at the point.

Reviewer #2 (Remarks to the Author):

In their revised manuscript the authors have addressed many of the raised concerns, at least in a textual, discussive approach. I, however, cannot suggest the manuscript for acceptance yet despite visible improvements. In some points I feel the authors have not comprehensively addressed the core of my concerns, which will need to be done.

In detail, I am (again) asking for the following (using the previous numbering) before this manuscript may be finally suggested for consideration:

1) On ATP and magnesium: Please add a Mg-titration for s2m and include the imino region (see also comment below). There are multiple studies that show a clear effect of Mg on s2m. The correct fold of s2m needs to be proven here, best by comparison of imino spectra to other studies.

2) ok

3) ok

4) ok

5+6) I am ok with sticking to s2m as the one readout. Still, for the reader and understanding of the ATP-effects over RNA-binding, please give known affinities for NTD and CTD with s2m (a reference is sufficient if this is known, or provide your own at least as a reasonable estimate). As by literature the affinity for CTD should be higher than for NTD?!

The manuscript did not and still does not (as far as I can see) state that s2m has been applied in its DNA-form! This is an important information, not only for the reproducibility of data! While I can understand the technical and economic reasons for this, the nature of the nucleic acid element matters a lot here. s2m RNA is a known to be a labile, bulged stem-loop and by no means we can expect to fold DNA in the same way (instead of a incomplete double strand e.g.). This is an essential part of the study and should not be ignored, especially as authors claim this study to provide a starting point for drug design. In line with comment 1 authors should provide proof for the folding of their DNA s2m and compare it to existing RNA spectra. Otherwise the rationale of choosing s2m based on its description as an NTD/CTD-binder before does not seem plausible to me. One could then just

pick a less complex ssRNA or any nucleic acid that should likewise bind to NTD and/or CTD. In any case, this information needs to be given in the manuscript as readers are currently left alone with not knowing the NA nature of this study.

7) Again on the specificity: I am fine with the explanation given and know what is meant. I still find the word specificity to appear to over-exploited in the text. See e.g. new page 5, line 125 ("interactions with S2m, a 32-mer specific stem-loop II nucleic acid motif"), where I do not understand the word specific at all.

For the protein side, certainly, there may be particular residues involved and others not so much. Hydrophobic interactions though will also lead to more effects in hydrophobic residues than in polar ones. But it is misleading to speak of s2m specifically binding to IDRs, NTD and CTD while no other RNA/DNA has been used. Maybe this is a matter of phrasing and can simply be written with more cautiousness. Also, the authors should briefly comment on why only particular Args in IDRs are binding to ligands and others not?

8) ok

9) ok, see #5+6 for the estimate of affinities (which are given as μ molar in the authors' response)

10) I do not see extensive improvements of figures except for shifting two of them to the SI. E.g., I still think a zoom-in for CSP during titration as basis of the K_d will be helpful. Full-spectral overlays are not easy to follow, also based on the color code.

11) No extensive text cuts became obvious to me. Certainly, new text had to be added for the revised version. The shortenings are though difficult to follow without any further indication. To shorten is just a suggestion I have, but if so, authors may help in guiding reviewers through extensive changes.

12) ok

For minor points, it seems that some of the issues have not been addressed although stated as changed by the authors:

- inconsistency in labeling in Fig. 1: NTD 40-180 vs. legend NTD 44-180
- page 3, line 79: I assume it shall be Fig. 1A and not S1A?
- line 102: ...binding to pockets on its NTD...? ATP's pockets? Please rephrase.
- page 23, line 528: were expressed
- Are the blue boxes in Figures 3-7 necessary?
- Suppl. Table: I assume K_d is meant as constant (if not rate is meant, which would be k_d)!

Reviewer #3 (Remarks to the Author):

In the revised version of the manuscript "Mechanism for ATP and nucleic acid to interplay in modulating LLPS of SARS-CoV-2 N protein," Mei Dang et al. have addressed comments and criticism by all three reviewers. Their presentation has improved, and over interpretation of data has been taken care of. I found the revised version quite satisfactory.

1. Though authors have discussed the role of ATP in the viral life cycle, it did appear too convincingly to me.

2. Discussion on the usage of S2m is still not appropriate.

3. Authors proposed binding of ATP in a specific manner to the N-term part of the protein; however, HSQC spectra suggested a wide range of binding. I do not think specific binding is the correct term.

Overall, the manuscript has improved, and I recommend it be accepted.

Point-to-point Response

Reviewer #1

1) Although, they have not been able to show some of the comparison 1H-13C spectra, they have been able to add a new 1H-15N spectra. But, this makes it a little bit more confusing to compare and understand the actual interactions.

Response: thanks so much for the kind comment.

In fact, previously due to the novelty of the ATP-binding at mM, we have assessed the binding effect of ATP on several ¹⁵N-/¹³C-labeled FUS and TDP-43 domains, and only found very minor shifts for ¹³C resonances. This is reasonable as for the new category of the ATP-binding we studied here, the *k_d* value is >mM. As a consequence, such a binding should have very weak effect on ¹³C chemical shifts. In fact, as seen in Fig. S3A, even the proton resonances of methyl group only have slight shifts. This is likely one reason why the ATP-binding to proteins at mM has been ignored until very recently.

2) On the concern about stoichiometry, authors gave a previous reference which does not answer the concerns. It's all about a model which can replicate the cellular/physiological concentration or ratio that works between each other. This is still a loophole. Some cellular work could have answered the combinational effect.

Response: thanks so much for kindly pointing out this and we really apologize for the previous misunderstanding.

We thought that the reviewer asked the question how the ratio of ATP and N protein alters at different stages of the infection.

For the stoichiometry of the ATP binding, there are different scenarios: for the well-folded nucleic-acid-binding domain, the stoichiometry is 1:1 as we previously characterized on a variety of nucleic-acid-binding domains, particularly the FUS RRM binding domain on which we have systematically titrated it with ATP, ADP, AMP, Adenine and Triphosphate (Ref. 29). On the other hand, for the well-folded proteins without the nucleic-acid-binding capacity, ATP only has very weak, non-specific perturbation as we showed on a control protein Profilin-1 in the same study (Ref. 29), as well as recently demonstrated by a Japanese group on a list of proteins (newly-added Ref. 58).

For the ATP binding to Arg residues within IDRs, we previously proposed that the Adenine and triphosphate can separately bind the side chains of Arg residues, thus establishing the bivalent binding (Ref. 28). Indeed, during the review process of the current manuscript, a theoretic study has been published by combining quantum mechanical method, mean-field theory and molecular simulations (Ren et al. [2022] *Sci Adv.* 8:eabo7885), revealing that the aromatic rings of ATP and triphosphate group indeed separately interact with Arg residues to induce and then dissolve LLPS of FUS as we experimentally observed before.

In the revised manuscript, we have added two new references and discussion on pages 19-20 as “Furthermore, our results unveiled that distinctive scenarios exist for the binding of ATP and nucleic acid to the folded domains such as NTD/CTD and IDRs. Briefly, for the folded domains with no capacity in binding nucleic acids, ATP only has very weak and non-specific interaction with them, as we previously observed on human profilin-1 (29) and the other group showed on a list of proteins (58). By contrast, for the nucleic-acid-binding domains such as FUS RRM domain (29) and NTD/CTD of SARS-CoV-2 N protein, ATP binds to the pockets within their nucleic-acid-binding interface, which is formed only

upon the correct folding by diverse residues located over the entire sequence. On the other hand for IDRs, ATP and nucleic acid appear to mainly bind Arg/Lys residues, as we experimentally found on FUS (28) and TDP-43 PLD (35) as well as very recently uncovered by combining quantum mechanical method, mean-field theory and molecular simulations (59).

3) Author mentioned the corresponding references (which were requested to be added) in the rebuttal letter. Authors should make sure that they have also added those in the manuscript.

Response: thanks so much for the kind comment.

We have added all references except for those about the *in vivo* change of ATP-related pathways/proteins during the infection as this topic is much beyond the focus of our current biophysical study. In particular, the second reviewer strongly required to further shorten the manuscript.

4) On the question of how did they measure viscosity change, the response was not at the point.

Response: thanks so much for the kind comment.

Really apologize for the confusion. As the viscosity we previously discussed is not that of the bulk solvent, but the viscosity of several layers of solution molecules surrounding ATP and proteins which were previously proposed in literatures. For this, it is almost impossible to experimentally measure it. So to avoid the confusion, we removed this speculation.

Reviewer #2

1) On ATP and magnesium: Please add a Mg-titration for s2m and include the imino region (see also comment below). There are multiple studies that show a clear effect of Mg on s2m. The correct fold of s2m needs to be proven here, best by comparison of imino spectra to other studies.

Response: thanks so much for the kind comment.

The result was presented in Fig. S5C and we would really apologize for not labelling Fig. S5C properly.

As my group has studied the interaction of ATP/nucleic-acid with various folded and unfolded proteins including TDP-43, FUS, Profilin-1, hSOD1, Crystallin and SARS-CoV-2 N protein etc, for each new target protein, we always conducted control titrations of Mg into protein and nucleic acid. With regard to the different domains of SARS-CoV-2 N protein, we have also done this set of NMR experiments and found no significant interactions of Mg with these domains and S2m. Previously we have presented these results to the reviewers for two papers with S2m (Dang and Song [2021] *QRB Discovery*, 2, 1; Dang and Song [2022] *Protein Sci.* 31, 345).

However, during the review of our paper in *Protein Science*, one reviewer requested to use the exact the same ATP-Mg complex to titrate S2m, as the reviewer commented that it is the ATP-Mg complex we used in all experiments and therefore only the ATP-Mg complex is the correct control.

Fig. S5C is the titration results of the ATP-Mg complex with S2m and from the results, there is no significant changes of S2m with the concentrations of the ATP-Mg complex even reaching up to 10 mM. In the revised manuscript, we have corrected the label of Fig. S5C.

5+6) I am ok with sticking to s2m as the one readout. Still, for the reader and understanding of the ATP-effects over RNA-binding, please give known affinities for NTD and CTD with s2m (a reference is sufficient if this is known, or provide your own at least as a reasonable estimate). As by literature the affinity for CTD should be higher than for NTD?!

The manuscript did not and still does not (as far as I can see) state that s2m has been applied in its DNA-form! This is an important information, not only for the reproducibility of data! While I can understand the technical and economic reasons for this, the nature of the nucleic acid element matters a lot here. s2m RNA is a known to be a labile, bulged stem-loop and by no means we can expect to fold DNA in the same way (instead of a incomplete double strand e.g.). This is an essential part of the study and should not be ignored, especially as authors claim this study to provide a starting point for drug design. In line with comment 1 authors should provide proof for the folding of their DNA s2m and compare it to existing RNA spectra. Otherwise the rationale of choosing s2m based on its description as an NTD/CTD-binder before does not seem plausible to me. One could then just pick a less complex ssRNA or any nucleic acid that should likewise bind to NTD and/or CTD. In any case, this information needs to be given in the manuscript as readers are currently left alone with not knowing the NA nature of this study.

Response: thanks so much for the kind comments.

In study of protein-nucleic acid interactions, it is challenging sometimes to precisely measure their binding affinity particularly when nucleic acids contain extra regions other than the exact binding sequence. My lab has projects focused on the binding of various nucleic acids with proteins such as RRM domains of TDP-43 and FUS. Previously we have

performed ITC titrations on them but none of them generated high-quality data for fitting out binding parameters. For NMR titrations, most nucleic acid sequences would lead to severe broadening and then disappearance of NMR peaks likely because k_d values are $\sim\mu\text{M}$.

In the paper we cited (Chen et al. [2007] *J Mol Biol.* 368:1075. Ref. 37), three forms of S2m of SARS-CoV-1 (RNA, ssDNA and dsDNA) have been studied. One rationale is that even for SARS-CoV-1, scientists found that its sub-genome RNAs could be reverse-transcribed into ssDNA or even dsDNA forms in human cells. For SARS-CoV-2, it has been recently shown that its genome RNAs not only could be reverse-transcribed into ssDNA, but also integrated into human genome (Zhang et al. [2021] Reverse-transcribed SARS-CoV-2 RNA can integrate into the genome of cultured human cells and can be expressed in patient-derived tissues. *PNAS.* 118,e2105968118). In this context, SARS-CoV-2 N protein is most likely to interact also with the DNA forms as N protein is highly expressed in the infected cells and involved these processes. However, due to the challenge, no precise binding affinity could be measured in Ref. 37.

For ssDNA form of S2m, we have already published two papers and previously conducted systematic NMR titrations of S2m into NTD and CTD. Unfortunately the majority of HSQC peaks disappeared at high S2m concentrations and only several peaks retained with shifts. Based on the shifted residues, k_d for binding NTD was fitted to be $\sim 30\ \mu\text{M}$ while for CTD is $\sim 10\ \mu\text{M}$. However, as the residues with remaining HSQC peaks might not be directly involved in binding, we did not include the data to avoid misleading.

As the reviewer kindly pointed out, in the current study, we only aimed to use S2m as a “readout/reporter” to reveal that ATP not only can bind NTD/CTD and Arg within IDRs, but is also able to displace S2m from binding NTD/CTD/IDRs. In this context, the exact affinity of S2m is not our focus as in any sense N protein *in vivo* is expected to bind a variety of viral and human DNA/RNA of variable sequences, lengths and secondary structures in cells. In particular, all these bindings are physiologically or pathologically relevant. In this regard, with S2m, our current results are sufficient to illustrate ATP and nucleic acids have the overlapped binding sites.

Currently, no biophysical study can completely reproduce the exact situation in cells. Therefore our current biophysical study only aimed to explore the fundamental interactions/principles of LLPD of N protein as modulated by ATP and nucleic acid. In this context, although nucleic acids of different types, sequences, lengths and secondary structures might result in differential affinity, it is anticipated not to change the fundamental interactions/principles. This is why at the very beginning we selected non-specific sequences such as A24 and A32. Strikingly, previous studies by other groups we cited on LLPS of SARS-CoV-2 N protein have also shown that the fundamental properties of LLPS remain the same regardless of being induced by RNA/DNA of variable sequences, lengths and secondary structures. Nevertheless, later we decided to use S2m as some reviewers for our previous papers strongly recommended.

Although in the current project, to assess the effect of nucleic acids of different sequences and lengths is much beyond our focus, we have experimentally addressed this fundamentally-critical issue by using TDP-43 prion-like domain (PLD). The results reveal that different types and sequences of nucleic acids have no detectable difference for the affinity in binding Arg/Lys within TDP-43 PLD. The manuscript was just scientifically accepted by *Communications Biology* and thus attached.

7) Again on the specificity: I am fine with the explanation given and know what is meant. I still find the word specificity to appear to over-exploited in the text. See e.g. new page 5, line 125 (“interactions with S2m, a 32-mer specific stem-loop II nucleic acid motif”), where I do not understand the word specific at all.

For the protein side, certainly, there may be particular residues involved and others not so much. Hydrophobic interactions though will also lead to more effects in hydrophobic residues than in polar ones. But it is misleading to speak of s2m specifically binding to IDRs, NTD and CTD while no other RNA/DNA has been used. Maybe this is a matter of phrasing and can simply be written with more cautiousness. Also, the authors should briefly comment on why only particular Args in IDRs are binding to ligands and others not?

Response: thanks so much for the kind comment.

Yes, as we presented above to address the comment by the first reviewer, the binding of ATP and ssDNA to N proteins, has different scenarios which are different from the classic “specific” binding. So in the revised manuscript, we removed “specific”.

Furthermore, to address this comment, a detailed discussion was developed and presented on pages 19-20 of the revised manuscript as “Furthermore, our results unveiled that distinctive scenarios exist for the binding of ATP and nucleic acid to the folded domains such as NTD/CTD and IDRs. Briefly, for the folded domains with no capacity in binding nucleic acids, ATP only has very weak and non-specific interaction with them, as we previously observed on human profilin-1 (29) and the other group showed on a list of proteins (58). By contrast, for the nucleic-acid-binding domains such as FUS RRM domain (29) and NTD/CTD of SARS-CoV-2 N protein, ATP binds to the pockets within their nucleic-acid-binding interface, which is formed only upon the correct folding by diverse residues located over the entire sequence. On the other hand for IDRs, ATP and nucleic acid appear to mainly bind Arg/Lys residues, as we experimentally found on FUS (28) and TDP-43 PLD (35) as well as very recently uncovered by combining quantum mechanical method, mean-field theory and molecular simulations (59).

10) I do not see extensive improvements of figures except for shifting two of them to the SI. E.g., I still think a zoom-in for CSP during titration as basis of the Kd will be helpful. Full-spectral overlays are not easy to follow, also based on the color code.

Response: thanks so much for the kind comment.

Previously we have published the results/figures for the binding of ATP to the isolated NTD and CTD. Most importantly, in the decision letter, the editor particularly requested to reduce the similarity in both presentation/discussion to our previous manuscripts. To achieve the goal, we had to move these figures to Supplementary Materials.

11) No extensive text cuts became obvious to me. Certainly, new text had to be added for the revised version. The shortenings are though difficult to follow without any further indication. To shorten is just a suggestion I have, but if so, authors may help in guiding reviewers through extensive changes.

Response: thanks so much for the kind comment.

In the previous revision we have deleted many discussions. However, as the reviewer could kindly see, three reviewers have a long list of comments and suggestions. To address them we had to add new data/description/discussion.

In the current revision, we further shortened the manuscript. However, we also needed to add more discussion. In this regard, the total length of the revised manuscript could possibly become unavoidably even longer.

For minor points, it seems that some of the issues have not been addressed although stated as changed by the authors:

- inconsistency in labeling in Fig. 1: NTD 40-180 vs. legend NTD 44-180
- page 3, line 79: I assume it shall be Fig. 1A and not S1A?

- line 102: ...binding to pockets on its NTD...? ATP's pockets? Please rephrase.
- page 23, line 528: were expressed
- Are the blue boxes in Figures 3-7 necessary?
- Suppl. Table: I assume K_d is meant as constant (if not rate is meant, which would be k_d)!

Response: thanks so much for kindly pointing out these errors and we have correct them.

Reviewer #3

1. Though authors have discussed the role of ATP in the viral life cycle, it did appear too convincingly to me.

Response: thanks so much for the kind comment.

At the very beginning, the finding of the role of ATP in the viral life cycle was also surprising to us. However, after we conducted more studies on other human protein systems include TDP-43 and FUS, we realized that ATP is a general modulator for protein-nucleic-acid interactions in all living cells as discussed in our another manuscript attached. As such, the life cycle of SARS-CoV-2 is avoidably modulated by ATP but amazingly SARS-CoV-2 appears to take this advantage to hijack into this modulating network for its replication.

2. Discussion on the usage of S2m is still not appropriate.

Response: thanks so much for the kind comment.

As the reviewer could kindly see, above we have addressed the similar comment by the second reviewer. Most importantly, we now attached our another manuscript on TDP-43 PLD, which is aimed to systematically address this fundamental question.

3. Authors proposed binding of ATP in a specific manner to the N-term part of the protein; however, HSQC spectra suggested a wide range of binding. I do not think specific binding is the correct term.

Overall, the manuscript has improved, and I recommend it be accepted.

Response: thanks so much for the kind comment.

We realized that this is a common question by all three reviewers. We have removed “specific” in the revised manuscript. Furthermore, to address this, a detailed discussion was developed and presented on pages 19-20 of the revised manuscript as “Furthermore, our results unveiled that distinctive scenarios exist for the binding of ATP and nucleic acid to the folded domains such as NTD/CTD and IDRs. Briefly, for the folded domains with no capacity in binding nucleic acids, ATP only has very weak and non-specific interaction with them, as we previously observed on human profilin-1 (29) and the other group showed on a list of proteins (58). By contrast, for the nucleic-acid-binding domains such as FUS RRM domain (29) and NTD/CTD of SARS-CoV-2 N protein, ATP binds to the pockets within their nucleic-acid-binding interface, which is formed only upon the correct folding by diverse residues located over the entire sequence. On the other hand for IDRs, ATP and nucleic acid appear to mainly bind Arg/Lys residues, as we experimentally found on FUS (28) and TDP-43 PLD (35) as well as very recently uncovered by combining quantum mechanical method, mean-field theory and molecular simulations (59).

Reviewers' comments:

Reviewer #2 (Remarks to the Author):

After a second round of revision, the authors have taken additional steps to improve the manuscript and certainly tried to meet the concerns of all three reviewers, which seem to overlap in the most critical parts.

It appears though that the authors have ignored a number of clear suggestions and requirements, not only brought up by this reviewer.

As I do not see a large problem in filling this gap (seeing that the story as such seems clear) I again ask authors to do the following:

1) Please add imino-regions of 1D spectra comparing s2m alone and with ATP-Mg! The current display in Fig. S5C is not the imino region.

2) In line with that, I am not convinced of the way how authors treat s2m without any further comment that it is used as DNA. Please provide a clear comparison that the DNA folds as the RNA would. This must be added. Also, there is still no methodological comment on s2m as being a DNA (unless I miss it in the manuscript). This has to be included.

I do not need another justification of s2m as DNA, but a clear comparison that justifies its use. Else one can use any nucleic acid, but then the story goes into a different overall direction.

IF s2m as DNA and RNA have been showed to similarly act in N-binding before, including the nucleic acid perspective, then this information can be used. However, it needs to be fairly mentioned that s2m is a DNA, but its use is justified (at least). Else, one might interpret this lack as if there is a good reason to not provide the requested comparison.

Of minor impact:

The authors also still ignore my suggestion to comment on the specific binding of some of the Args in IDRs. I do not need far-reaching comparisons to other work here, but a statement that makes into the text.

Also, regarding my comment on the improvement of figures: I did not mean to get a justification for why figures have been moved, but simply noted that those are the only changes I saw. I am fine with that, but see large potential for clearer figures. Anyway, I would leave this to an editorial process, while the above points are mandatory before acceptance by me.

Detailed Changes in the revised manuscript.

To address the kind comments by the editor and reviewers, I have made the following revisions:

1. For the justification to use ssDNA version of S2m, I have added it into Introduction on page 5 (colored in blue in the version with changes tracked).
2. For the concerns on the quality of the spectra in figures, I have carefully checked the quality of these figures but found no problem in TIF format. Only today I suddenly realized that this might be due to the possibility that all figures the reviewer examined were already converted into pdf format and consequently their resolution became very lower. So as kindly requested, I have prepared two new zoom-in subfigures as Figure S4D and Figure S5D with their corresponding descriptions added into the revised manuscript (colored in blue in the version with changes tracked).

Point-to-point response

Editorial board

The suggestions from the reviewer are specific and friendly and I think these can be addressed by a quick revision.

From the editorial board, we request you to kindly take a close look at the figures and improve their quality. I also request that when preparing the next version you kindly refer to the following recommendation on your original version "On Figures: Figures should be designed/labeled (inset text) to express what is claimed. For some of them, I suggest providing additional zoom-ins rather than sole full-spectral views. This includes the Kd determination: Exemplary shifting residues should be shown indicating the data quality (seeing the minor CSPs present at maximal ATP concentration). In its present form the large number of HSQCs hardly makes a link to the derived data."

Response: Many thanks to the kind instructions.

As you can kindly see below, to adequately address the remaining comments by the second reviewer, I have:

1) checked the resolution of all figures and realized that the figures in pdf have the much lower resolution than the original ones in tif. For the main figures, the tif figures were already provided and will be published. For the supplemental figures, I now also provided a separate file other than pdf file of Supplemental Materials with the original figures in tif embedded in word.

2) I have provided the expanded shift tracings as the reviewer kindly suggested.

3) I have added the detailed discussions with new references into the main text to address the remaining comments as detailed below. The revised parts were coloured in blue in the revised manuscript with changes tracked.

Reviewer #2

1) Please add imino-regions of 1D spectra comparing s2m alone and with ATP-Mg! The current display in Fig. S5C is not the imino region.

Response: Many thanks to the kind instruction.

First I would apologize for this. Yes, Fig. S5C is the one we already used as supplementary figure to address the same comment by one reviewer for our previous paper (Dang & Song. [2021] CTD of SARS-CoV-2 N protein is a cryptic domain for binding ATP and nucleic acid that interplay in modulating phase separation. *Protein Science*. 31,345). That reviewer particularly requested to collect the 1D region for the protons of the base aromatic rings and NH₂ which are directly involved in base pairing if any doubled stranded helix is formed. Therefore, if ATP-Mg complex is able to trigger any conformational alternation, the chemical shifts of these protons will for sure shifted. In other words, if the chemical shifts of these protons are not shifted, the result will provide unambiguous evidence that no conformational alteration occur for S2m upon addition of ATP-Mg complex.

Due to the catastrophic pandemic, my lab was completely ruined and now I have no staff in the lab. The first author of the current manuscript who was my previous PhD student and supposed to do postdoc in my lab already left Singapore and will not be back due to the heart damage by vaccine. The second author who did Hons project in my lab and supposed to continue the PhD study suddenly decided not to continue academic career. Most seriously, due to the very small researcher pool in Singapore, currently I am unable to recruit any qualified staff to do NMR. So I need to do everything myself but I am currently on leave and can only access the previous data in my mobile disks.

So I tried hard to reach the first author to get the raw NMR data for the ATP-Mg titrations but failed. Fortunately I eventually found the processed 1D spectra and prepared the new Fig. as Fig. S5D. Although previously for whatever nucleic acids, I always asked to collect 1D spectra down to 20 ppm, this one only goes down to 11.5 ppm. I am not sure this is due to the processing script or not. Fortunately, for this control experiment required for the current story, the obtained results are sufficient to unambiguously indicate that no conformational alteration occurred for S2m upon addition of ATP-Mg complex.

I agree that the NMR signals of the labile protons contain a rich set of information particularly for the dynamics. On the other hand, as they constantly exchange with water, they are significantly affected by the solution conditions such as ionic strength. So it is very challenging to use this region to reach very certain conclusion without exhaustive studies. So I wish to study these issues in the future.

To address this comment, I have added Fig. S5D and corresponding discussion on page 15 as "To exclude the possibility that the above observation is due to the alteration of the conformation of S2m by ATP-Mg complex, we titrated ATP-Mg complex into a S2m sample as monitored by 1D proton NMR spectroscopy on the protons of the base aromatic rings and NH₂ which are directly involved in base pairing. The obtained results showed that addition of ATP-Mg complex even up to 10 mM triggered no shift of these NMR signals (Fig. S5C and S5D), thus unambiguously indicating that no conformational alteration occurred for S2m upon addition of ATP-Mg complex."

2) In line with that, I am not convinced of the way how authors treat s2m without any further comment that it is used as DNA. Please provide a clear comparison that the DNA folds as the RNA would. This must be added. Also, there is still no methodological comment on s2m as being a DNA (unless I miss it in the manuscript). This has to be included.

Response: Many thanks to the kind comment.

Actually I started to work on SARS-CoV-1 since 2003 and knew that its genome RNA could be reverse-transcribed into ssDNA as well as integrated into human genome as dsDNA. This is why the group in Taiwan studied the interactions between N protein and RNA, ssDNA and dsDNA and found they bind in the same mode (Ref. 37). Now this has been confirmed on SARS-CoV-2 by MIT group and published in PNAS. So ssDNA form of S2m is also a biological ligand of N protein.

To address this issue, I have added a paragraph with several new references on pages 5-6 as “Previously RNA, ssDNA and dsDNA forms of SARS-CoV-1 S2m have been shown to bind N protein in the same mode (37). Furthermore, for SARS-CoV-2, it has been very recently shown that its genome RNA not only could be reverse-transcribed into ssDNA, but also integrated into human genome as double-stranded DNA (dsDNA) (38). This finding suggests that the different forms of S2m may be the biological ligands of the N protein, which is in general consistent with the notion that many nucleic-acid-binding domains have a very broad spectrum of specificity (28-31,39-42). For example, human TDP-43 protein containing RNA-binding motif (RRM) domains has been uncovered to functions by binding a large array of RNA and DNA including more than 6000 RNA species of diverse sequences and structures (40-42). As such, here we used the ssDNA form of S2m because unlike RNA which is prone to degradation by RNase extensively existing in environments, ssDNA has a very high chemical stability, thus allowing to acquire time-consuming NMR spectra (41,42).”

Of minor impact:

1) The authors also still ignore my suggestion to comment on the specific binding of some of the Args in IDRs. I do not need far-reaching comparisons to other work here, but a statement that makes into the text.

Response: Many thanks to the kind comment.

I have enhanced the discussion on this point by revising the discussion with new references on page 20 as “On the other hand, for IDRs ATP and nucleic acid were now shown to bind Arg/Lys residues in the residue-specific manner by establishing electrostatic interactions between triphosphate group of ATP or phosphate group of nucleic acid and side chain cations of Arg/Lys as well as π - π / π -cation interactions between base aromatic rings and Arg/Lys side chains, as we experimentally found on FUS (28) and TDP-43 PLD (35,41) as well as very recently uncovered by combining quantum mechanical method, mean-field theory and molecular simulations (63).”

2) Also, regarding my comment on the improvement of figures: I did not mean to get a justification for why figures have been moved, but simply noted that those are the only changes I saw. I am fine with that, but see large potential for clearer figures. Anyway, I

would leave this to an editorial process, while the above points are mandatory before acceptance by me.

Response: Many thanks to the kind instruction.

As kindly instructed, I have added the expanded tracings as Fig. S4D and corresponding description on page 9 as “Fig. S4D presents the expanded shift tracings of two selected residues: Ala90 within NTD and Arg32 from IDR1.”